# TASK-AWARE MODEL MERGING VIA FISHER-WEIGHTED MEDIAN

## ABSTRACT

Fine-tuning large language models (LLMs) on task-specific data provides strong
in-domain performance but limits generalization and requires storage of many
specialized models. Retraining a unified multitask model is often infeasible, as
it demands task-specific training data that may be unavailable, raise privacy con-
cerns, or incur prohibitive computational costs. Model merging has been proposed
as an alternative solution that effectively integrates the distinct strengths of sev-
eral fine-tuned models into a single, comprehensive model. The majority of model
merging approaches rely on performing arithmetic operations directly on model
parameters. Although research in model merging has expanded significantly in re-
cent years, two distinct approaches have become dominant: 1) techniques that mit-
igate interference from redundant parameters and sign conflicts, and 2) techniques
that account for the varying sensitivity of individual parameters. However, these
two approaches operate independently without considering each other's strengths
and remain disconnected from each other. In this work, we aim to unify these
two well-established yet currently disconnected approaches by integrating insights
from both the approaches. We propose `DRIFT-MEDIAN`, a unified framework for
merging models that leverages *Fisher information* to assign appropriate weights
to the task vectors. Our contribution lies in the development of a closed-form so-
lution of loss function grounded in the Fisher-weighted median. The formulation
ensures that parameter contributions reflect both sensitivity and relevance, lead-
ing to more robust model merging. This mechanism prioritizes parameters with
high task-specific sensitivity in the merged representation, while naturally dimin-
ishing the influence of less important parameters. Comprehensive experiments on
Llama-3.1-8B, Llama-3.2-3B, Llama-2-7b, GPT-2, CLIP-ViT-B/32 models across
mathematics, coding, multilingual reasoning, safety, instruction following, GLUE
benchmark and vision tasks demonstrate that `DRIFT-MEDIAN` outperforms ex-
isting model merging methods.

## 1 INTRODUCTION

Large Language Models (LLMs) (Radford et al., 2019; Grattafiori et al., 2024; Touvron et al., 2023)
usually require fine-tuning on domain-specific datasets to achieve optimal performance in special-
ized tasks. Although this approach yields strong in-domain performance, it introduces significant
practical challenges in terms of substantial storage, computational costs, and limited data availability
or data privacy constraints. Model merging has emerged as a compelling solution to these challenges
by combining parameters from independently fine-tuned models of identical architecture into a sin-
gle unified model (Ilharco et al., 2022a; Hinton et al., 2006; Yadav et al., 2023; Yu et al., 2024; Yang
et al., 2023), eliminating the need for costly retraining. Existing approaches operate either in the pa-
rameter space (PS), where merging directly manipulates model weights (Jin et al., 2023; Shoemake,
1985; Akiba et al., 2025; Yang et al., 2023), or in data-flow space (DFS), where individual model pa-
rameters remain intact while optimization focuses on inference pathways (Kim et al., 2024). Hybrid
approaches such as Evolutionary Model Merging (Akiba et al., 2025) incorporate elements of both
paradigms. Despite significant progress in this field, there is still considerable scope for improving
the effectiveness of current parameter-space merging methods.

**Challenges and Motivation:** A comprehensive review of current parameter-space merging tech-
niques highlights two distinct approaches. While each addresses complementary facets of the merg-

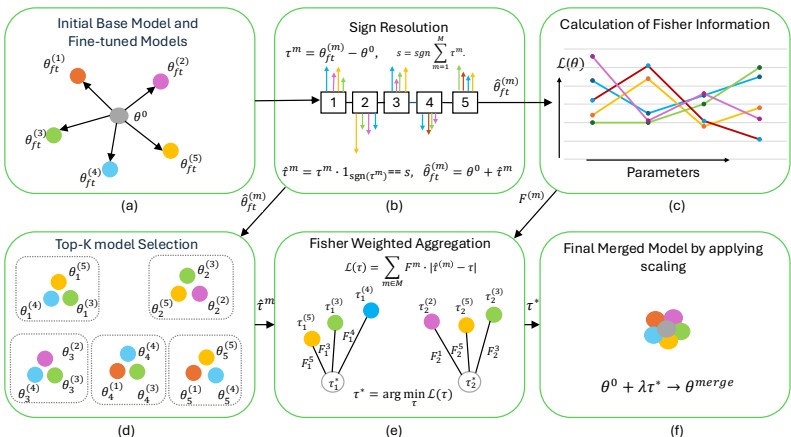

Figure 1: Overview of the steps involved in the proposed model merging approach. (a) $\theta^0$ and $\theta_{ft}^{(m)}$ (with specific color) represent base model and different fine-tuned models respectively. (b) In the sign resolution step, task vectors that agree in sign are retained to compute the fine-tuned parameters $\hat{\theta}_{ft}^{(m)}$. 1, 2, 3, 4, 5 inside small square boxes represent parameter indices. (c) Diagonal Fisher matrix is estimated from the fine-tuned parameters. (d) Top-$K$ models are selected at each coordinate based on distance from the base model, using a specified threshold value. Superscript and subscript of $\theta$ represent respective fine-tuned model and parameter indices. (e) Finally, Fisher-weighted aggregation yields $\tau^*$, followed by scaling (f) to obtain the merged parameters $\theta^{merge}$.

ing challenge, they fall short of exploiting the synergistic benefits that their integration could offer - (1) approaches that resolve parameter interference during merging but do not consider parameter sensitivity to task performance (Yadav et al., 2023; Yu et al., 2024), and (2) methods that account for individual parameter sensitivity but do not address parameter interference aspects during the merging process (Matena & Raffel, 2022).

An example of the first type is TIES merging Yadav et al. (2023) that tackles two key sources of interference namely the sign disagreement, where task vectors exhibit opposing directional updates that cancel each other during averaging, and redundant parameters, where uninformative updates dilute the contributions of more significant changes. However, it does not consider individual parameter sensitivity to task performance, potentially allowing less critical parameters to overshadow more influential ones during the merging process.

An alternative approach, represented by Fisher merging (Matena & Raffel, 2022), incorporates parameter sensitivity through Fisher information weighting. It tackles the problem as maximizing the joint likelihood of the models' posterior distribution over parameters, demonstrating that parameter averaging is equivalent to employing an isotropic Gaussian distribution as an approximation for the posterior in each model. The merged parameter is estimated via weighted averaging over parameter's Fisher information. However, it does not address parameter interference issues, resulting in redundant parameters contributing to task conflicts and computational overhead during inference.

This separation suggests a potential opportunity, as we hypothesize that effective model merging inherently requires both: (a) accounting for parameter sensitivity across individual models to preserve critical task-specific knowledge, and (b) appropriately managing parameter interference to eliminate redundancy.

**Overview and Contributions**: Motivated by aforementioned observations, we propose a method named `DRIFT-MEDIAN`, a parameter-space merging framework that unifies insights from both interference reduction and sensitivity-based merging techniques.

`DRIFT-MEDIAN` operates through a carefully chosen sequence of operations, as illustrated in Figure 1. We first compute task vectors (Ilharco et al., 2023) by subtracting base model parameters from fine-tuned parameters, then perform sign resolution to eliminate directionally conflicting updates by establishing consensus directions at each coordinate. To quantify parameter importance, we

compute empirical Fisher information matrices that capture the sensitivity of each parameter to task performance. Unlike prior methods that perform pruning within individual models, we introduce coordinate-wise Top-$K$ selection that operates across models, retaining only the most informative task vectors at each parameter position to prevent both parameter crowding and scarcity issues.

The core idea lies in our Fisher-weighted median aggregation, which we formulate as an $L_1$-minimization problem with a closed-form solution based on the Fisher-weighted median (Gurwitz, 1990). This approach ensures that parameter contributions reflect both sensitivity and relevance while maintaining robustness to outliers which gives a critical advantage over traditional Fisher-weighted averaging approaches that can be dominated by extreme values. Through extensive ablation studies, we confirm the importance of each component in our framework, establishing `DRIFT-MEDIAN` as a robust and principled approach to parameter-space model merging. Key contributions of our proposed framework are as follows:

- **Fisher-weighted median aggregation:** We propose `DRIFT-MEDIAN`, a parameter- space merging method where we introduce a closed-form solution, based on the Fisher-weighted median, ensuring that parameter contributions reflect both sensitivity and relevance, leading to robust and balanced merging.

- **Coordinate-wise Top-$K$ selection:** Unlike prior methods that prune updates within each model independently, we perform cross-model coordinate-wise filtering to retain only the most informative task vectors, reducing noise and ensuring balanced aggregation.

- **Comprehensive evaluation:** Experiments across mathematics, coding, multilingual reasoning, safety, vision tasks and instruction-following demonstrate that `DRIFT-MEDIAN` consistently achieves higher performance retain rate (PRR) compared to prior methods, while ablation studies confirm the importance of each design choice.

## 2 RELATED WORK

**Background:** Given a base model with parameters $\theta^{(0)}$ and a collection of fine-tuned models $\{\theta^{(m)}\}_{m=1}^M$ specialized for tasks $\{t_1, t_2, \ldots, t_M\}$, the objective is to consolidate these specialized weights into a unified multitask model that maintains strong performance across all constituent domains. The central concept underlying parameter-space merging is the task vector, formally defined as $\tau^{(m)} = \theta^{(m)} - \theta^{(0)}$, where $\theta^{(0)}$ represents the base model parameters and $\theta^{(m)}$ denotes the parameters of a model fine-tuned for task $m$. Task Arithmetic (Ilharco et al., 2023) demonstrated that these task vectors effectively encode task-specific knowledge and that their addition to the base model successfully transfers the corresponding task capabilities. However, naive averaging of task vectors often leads to destructive interference with conflicting parameter directions, prompting the development of sophisticated merging techniques.

**Parameter Interpolation and Weight Averaging Methods**: Early approaches to model merging focused on linear interpolation techniques, leveraging the observation that despite the inherent non-linearity of neural networks, linear combinations of their weights can preserve high accuracy when the constituent models share common optimization trajectories (Choshen et al., 2022; Ilharco et al., 2022b; Izmailov et al., 2019; Wang et al., 2024; Choi et al., 2024; Daheim et al., 2024). Choshen et al. (2022) proposed a straightforward weight averaging approach for fusing fine-tuned models, demonstrating superior performance compared to using pretrained models alone. Building on this foundation, Wortsman et al. (2022) introduced "model soups", where multiple models fine-tuned with different hyperparameters are combined through weight averaging rather than selecting the single best-performing model based on validation metrics. This approach consistently improves performance over individual model selection. Similar improvements through weight averaging have been reported by Ilharco et al. (2022b); Matena & Raffel (2022); Li et al. (2022). More sophisticated interpolation methods have emerged to address specific challenges in parameter combination. Jin et al. (2023) developed a technique for determining merged model parameters through closed-form solutions, formulating the problem as local linear regression for individual layers within the model. These methods call for advanced merging strategies but typically do not account for parameter importance or interference effects.

**Task Vector Arithmetic and Interference Resolution**: Ilharco et al. (2023) formalized the concept of task vectors and demonstrated their effectiveness in model editing through arithmetic operations.

However, this approach revealed fundamental challenges when task vectors exhibit conflicting update directions, leading to the development of interference-aware methods. TIES merging (Yadav et al., 2023) addresses these interference issues through two key innovations. First, a trim step retains only the largest parameter deviations at each coordinate, suppressing redundant or weak updates that dilute informative changes. Second, step ensures directional consistency by choosing the majority sign across models at each coordinate and zeroing out conflicting updates. While effective at reducing destructive interference, TIES merging does not incorporate parameter sensitivity considerations, potentially allowing less critical parameters to overshadow more influential ones during aggregation. DARE (Yu et al., 2024) employs a complementary strategy of randomly dropping delta parameters with probability $p$ and rescaling the remaining parameters to maintain overall magnitude. This stochastic approach provides regularization benefits but lacks principled parameter importance weighting. Recent work such as SCE-merging (Wan et al., 2025) uses variance and magnitude-based criteria together with sign-consistency rules to identify stable parameters across models, while PCB-merging (DU et al., 2024) focuses on balancing inter-model and intra-model competition among task vectors.

**Sensitivity-Aware, Domain-Specific and Sparse Model Fusion Methods** : Fisher merging (Matena & Raffel, 2022) formulates merging as maximizing the joint likelihood of models' posterior distributions over parameters, demonstrating that parameter averaging is equivalent to using isotropic Gaussian approximations for each model's posterior. This ensures that parameters with higher estimated importance exert stronger influence during merging. However, Fisher merging does not address parameter interference issues, allowing redundant parameters to contribute to task conflicts and increasing computational overhead as the number of models grows. Recent work has developed specialized merging techniques for specific application domains. Zhou et al. (2024) introduced model exclusive task arithmetic for billion-scale models, while Djuhera et al. (2025); Hammoud et al. (2024) focus on maintaining safety alignment during merging procedures. These domain-specific approaches highlight the importance of preserving critical model properties beyond task performance. LoRA merging methods (Shah et al., 2024; Shenaj et al., 2024; Stoica et al., 2025; Yin et al., 2025) are designed to handle the unique properties of low-rank parameter updates, which present different challenges compared to full parameter fine-tuning. Similarly, vision-specific merging techniques (Zhu et al., 2025) have been developed to address the particular characteristics of computer vision models.

## 3 PROPOSED METHOD

Suppose $\theta^{(0)} \in \mathbb{R}^N$, and $\{\theta^{(m)}\}_{m=1}^M$ denote the parameter vectors for the base model, and a collection of fine-tuned models derived from the same base model, respectively. We denote the corresponding task vectors as $\tau^{(m)} = \theta^{(m)} - \theta^{(0)}$, that capture the parameter displacements induced by fine-tuning. Our objective is to combine these task vectors into a single stable representation that preserves salient task information (Algorithm 1) while mitigating destructive interference. In our method, we accomplish this via the following steps - (i) performing *sign resolution* to eliminate directionally conflicting updates, (ii) computation of *Fisher information* to quantify the sensitivity of each parameter, (iii) applying *coordinate-wise Top-K filtering* to retain only the strongest displacements, (iv) *computing a merged task-vector* by applying Fisher-weighted coordinate-wise median across the filtered task vectors, and (v) *scaling* on the aggregated neurons to compensate for the lost neurons during merging. We describe all of these in the subsequent sections.

### 3.1 SIGN RESOLUTION

Consider a collection of task vectors $\{\tau^{(1)}, \ldots, \tau^{(M)}\}$, each defined over coordinates $i \in \{1, \ldots, d\}$. At a given coordinate $i$, the corresponding set of entries is denoted as $\{\tau_i^{(1)}, \ldots, \tau_i^{(M)}\}$. Since these entries may take both positive and negative values, averaging them directly can result in destructive elimination of directional consistency, thereby discarding potentially stable task-specific knowledge.

To ensure alignment, we inherit the concept of sign consensus from TIES merging (Yadav et al., 2023). This is determined by the sign of the aggregated update across tasks as $s_i = \text{sign}\left(\sum_{m=1}^M \tau_i^{(m)}\right)$. The consensus sign $s_i$ specifies the dominant orientation

of update at coordinate $i$. Contributions inconsistent with this orientation are suppressed via the pruning rule given as follows $\hat{\tau}_i^{(m)} = \begin{cases} \tau_i^{(m)}, & \text{if } \text{sign}(\tau_i^{(m)}) = s_i \\ 0, & \text{otherwise} \end{cases}$. The result-ing collection $\{\hat{\tau}^{(1)}, \ldots, \hat{\tau}^{(M)}\}$ is therefore sign-consistent by construction. At each coordinate $i$, only those updates that are aligned with the consensus direction are preserved, while conflicting contributions are eliminated. This procedure removes destructive interference and yields a representation in which all retained task information is coherently oriented.

### 3.2 SENSITIVITY ANALYSIS VIA DIAGONAL FISHER INFORMATION MATRIX

While directional alignment ensures that task updates no longer interfere destructively, it does not incorporate the relative importance of different parameters. Not all coordinates contribute equally to model behavior in that some parameters are highly sensitive and strongly influence the predictive distribution, whereas others are less critical. To account for this, we introduce an importance weighting scheme based on empirical Fisher information (Matena & Raffel, 2022). Formally, consider coordinate $i$ in a model $m$. We define its empirical Fisher information as

$$F_i^{(m)} = \mathbb{E}_{(x,y) \sim D_m}\left[\left(\frac{\partial}{\partial \hat{\theta}_i} \log p\left(y \mid x; \hat{\theta}_i^{(m)}\right)\right)^2\right], \quad (1)$$

where $D_m$ denotes the data distribution associated with task $m$, $\theta^{(0)}$ is the reference initialization point (e.g., the pre-trained parameters), and $\hat{\theta}_i^{(m)}$ is the sign-aligned model weight and $\log p\left(y \mid x; \hat{\theta}_i^{(m)}\right)$ represent corresponding model posterior. Intuitively, $F_i^{(m)}$ quantifies the sensitivity of the model's predictive likelihood with respect to perturbations in parameter $\theta_i$. A large value of $F_i^{(m)}$ indicates that even small changes in $\theta_i$ have a substantial effect on the likelihood, implying that this coordinate is of high functional importance for task $m$.

Conversely, a small Fisher value suggests that $\theta_i$ is relatively insensitive and thus less critical. Accordingly, when merging task vectors, the update contribution $\hat{\tau}_i^{(m)}$ should be weighted proportionally to its Fisher information. This ensures that parameters with high task-specific sensitivity exert stronger influence on the merged representation, while less important directions are naturally down-weighted. In combination with directional alignment, Fisher weighting therefore produces a merged update that is both sign-consistent and importance-aware, preserving critical task knowledge while suppressing noise from less informative coordinates. Following common practice, we use the diagonal of the Fisher matrix, which requires $\mathcal{O}|\theta|$ memory for storage. In contrast, storing the full Fisher matrix would require $\mathcal{O}|\theta|^2$ memory, making it impractical for large models.

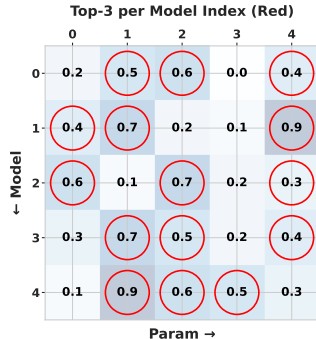

(a) Top-K Selection (TIES)

(b) Top-K Selection (Ours)

Figure 2: (a) In TIES Top-$K$ selection, only a single task vector at index 3, while four are concentrated at indices 1, 2, 4. (b) In contrast, our inter-model selection distributes task vectors more evenly, preventing crowding at certain parameter indices and scarcity at others, thereby ensuring consistent influence per index and avoiding dilution during aggregation.

### 3.3 COORDINATE-WISE TOP-$K$ SELECTION

After applying directional alignment and importance weighting, many parameters still exhibit small residual updates. These weak displacements are typically uninformative and can introduce noise into the merged representation. To mitigate this issue, we retain only the strongest task contributions on a per-parameter basis using an *inter-model* Top-$K$ selection strategy. Our motivation for this coordinate-wise approach follows the perspective of Qu & Horváth (2025), which argues that model merging fundamentally decomposes into a set of independent one-dimensional estimation

problems, one for each parameter. Consequently, interference, variance, and estimator instability arise *per coordinate*. In contrast, *intra-model* (model-wise) Top-$K$ performs sparsification independently within each model, ignoring how other models distribute their update mass. This often forces multiple task vectors to concentrate disproportionately large updates on a small subset of parameters, thereby increasing cross-task interference. Our inter-model Top-$K$ procedure avoids this issue by directly limiting how many models may influence any given coordinate, ensuring balanced competition across tasks precisely where the merged model must ultimately produce a single parameter value.

For task $m$ at coordinate $i$, define the update magnitude $d_i^{(m)} = |\hat{\tau}_i^{(m)}|$. Among the set $\{d_i^{(1)}, \ldots, d_i^{(M)}\}$, we select the $K$ largest values, with $K = \lfloor \kappa \cdot M \rfloor$, $\kappa \in (0, 1]$, where $\kappa$ denotes the 'keep-ratio'. Suppose $\delta_i = \min\left(\text{Top-}K\{d_i^{(1)}, \ldots, d_i^{(M)}\}\right)$, represent the cutoff magnitude for retention for coordinate $i$. Then the set of retained task indices is given by, $M_i = \{m \in \{1, \ldots, M\} : d_i^{(m)} \geq \delta_i\}$. Thus, at each coordinate, only those task updates with sufficiently large magnitude - specifically, the top fraction $\kappa$ - are preserved. This ensures that the merged representation emphasizes the most informative displacements while discarding weak or noisy contributions.

Consequently, our approach leads to a more consistent and conflict-free parameter aggregation. We illustrate this phenomenon in Figure 2, where TIES selects the Top-$K$ in an intra-model manner, whereas our method selects the Top-$K$ in an inter-model manner.

### 3.4 FISHER-WEIGHTED AGGREGATION

After directional alignment, importance weighting, and Top-$K$ filtering, we obtain a refined set of task-specific displacements at each coordinate. The final step is to merge these retained updates into a single consensus displacement. This can be formulated directly in parameter space as the solution of a *Fisher-weighted absolute-deviation minimization* problem. The objective can be expressed as:

$$L_i(\theta_i) = \sum_{m \in M_i} F_i^{(m)} \cdot \left|\hat{\theta}_i^{(m)} - \theta_i\right| \tag{2}$$

where $M_i$ is the set of retained task indices from the Top-$K$ filtering, and $F_i^{(m)}$ denotes the empirical Fisher information of parameter $i$ for task $m$. For coordinate $i$, let $\tau_i$ denote the candidate merged displacement. Since each task-specific parameter vector is expressed as $\hat{\theta}^{(m)} = \theta^{(0)} + \hat{\tau}^{(m)}$, the aggregation objective Equation 2 can be expressed in task-vector space as:

$$L_i(\tau_i) = \sum_{m \in M_i} F_i^{(m)} \cdot \left|\hat{\tau}_i^{(m)} - \tau_i\right| \tag{3}$$

This formulation enforces proximity between the merged displacement $\tau_i$ and the filtered task-specific values $\hat{\tau}_i^{(m)}$, while weighting each contribution according to its sensitivity. Parameters with larger Fisher values exert stronger influence, reflecting their higher functional importance. We adopt an $L_1$ (absolute-deviation) objective instead of the standard $L_2$ loss because the $L_1$ metric is more robust to outliers.

**Closed-form Fisher-weighted Median**

The minimizer of the Fisher-weighted absolute-deviation loss admits a closed-form characterization in terms of a weighted median. Specifically, the optimal merged displacement $\tau_i^*$ at coordinate $i$ is given by the Fisher-weighted median of the retained updates $\{\hat{\tau}_i^{(m)} : m \in M_i\}$. Formally, $\tau_i^*$ is defined as the value satisfying

$$\sum_{\hat{\tau}_i^{(m)} < \tau_i^*} F_i^{(m)} \leq \frac{1}{2} \sum_{m \in M_i} F_i^{(m)} \quad \text{and} \quad \sum_{\hat{\tau}_i^{(m)} > \tau_i^*} F_i^{(m)} \leq \frac{1}{2} \sum_{m \in M_i} F_i^{(m)}. \tag{4}$$

In other words, the Fisher weights of task-specific updates lying to the left and to the right of the solution $\tau_i^*$ each account for at most half of the total weight. This solution possesses a crucial robustness property: unlike Fisher-weighted means, which are highly sensitive to outliers due to squaring, the Fisher-weighted median ensures that extreme values cannot dominate the aggregate

Table 1: Results on GPT-2. The reported values correspond to absolute scores obtained on the validation set, since the test set is not publicly accessible.

| Method | COLA | MNLI | MRPC | QNLI | QQP | RTE | SST2 | Mean | $\overline{\text{PRR}}$ |
|---|---|---|---|---|---|---|---|---|---|
| Fine-tuned Models | 76.8 | 82.1 | 80.4 | 88.3 | 89.6 | 65.3 | 91.2 | 82.0 | - |
| Model Averaging | 55.0 | 55.1 | 51.0 | 57.6 | 76.7 | 44.8 | 52.5 | 56.1 | 68.5 |
| Task Arithmetic | 68.7 | 68.6 | 69.6 | 70.5 | 81.8 | 47.3 | **83.6** | 70.0 | 85.0 |
| TIES | 68.4 | 71.4 | 68.4 | 69.6 | 82.4 | 47.7 | 81.8 | 70.0 | 84.9 |
| Fisher Merging | 54.8 | 58.0 | 39.5 | 63.3 | 81.5 | 49.1 | 64.7 | 58.7 | 71.4 |
| Localize & Stitch | 64.1 | **76.1** | 48.0 | 65.5 | **83.1** | **53.1** | 55.7 | 63.7 | 77.9 |
| Ours | **69.1** | 71.3 | **70.1** | **83.0** | 79.3 | 50.2 | 77.3 | **71.5** | **86.9** |

Table 2: Results on Llama-3.1-8B models; The results are reported in relative percentage w.r.t. the fine-tuned models.

| Method | Maths | Multilingual | Instruction | Coding | Safety | $\overline{\text{PRR}}$ |
|---|---|---|---|---|---|---|
| Model Averaging (excl. embed) | 92.70 | 96.58 | 44.13 | 89.57 | 74.99 | 78.84 |
| Model Averaging (incl. embed) | 93.18 | 96.83 | 42.16 | 89.56 | 76.13 | 78.84 |
| Task Arithmetic | 93.85 | 91.80 | 56.02 | 90.92 | 79.63 | 82.85 |
| TIES | 96.44 | 95.95 | 51.53 | 90.40 | 83.91 | 83.75 |
| DARE | 91.90 | 89.84 | 54.31 | 87.35 | 77.55 | 80.60 |
| Fisher Merging | 89.17 | 96.53 | 61.19 | 87.37 | 88.46 | 83.07 |
| Localize & Stitch | **97.04** | **97.00** | 45.26 | 85.98 | 61.28 | 77.32 |
| Ours | 85.19 | 89.44 | **71.24** | **94.45** | **101.47** | **87.51** |

unless they are supported by sufficiently large Fisher weight. As a result, the merged displacement is both importance-aware and resistant to spurious task updates. We obtain the closed-form expression for the Fisher-weighted median following (Aho & Hopcroft, 1974; Blum et al., 1973; Gurwitz, 1990), and present the derivation in Appendix B.

**Scaling** Since sign pruning and Top-$K$ filtering reduce the effective magnitudes of task displacements, a rescaling step is applied to restore the overall adaptation strength (Ilharco et al., 2023; Yadav et al., 2023; Yu et al., 2024). For each coordinate, the merged displacement is given by $\tau_i^{\text{merge}} = \lambda \cdot \tau_i^*$, where $\lambda > 0$ is a global scaling factor. After all these steps, the final merged model is then constructed as follows: $\theta^{\text{merge}} = \theta^{(0)} + \tau^{\text{merge}}$. The scaling factor $\lambda$ serves as a tunable control that balances the contributions of the pre-trained model $\theta^{(0)}$ and the aggregated task updates. A larger value of $\lambda$ increases the influence of task-specific displacements, causing the merged model to drift further from the base model, while smaller values preserve closer adherence to the pre-trained initialization. Further discussion on the implementation details and design choices is provided in Appendix C.

# 4 EXPERIMENTAL RESULTS

## 4.1 EXPERIMENTAL SETUP

**Baseline Methods** We compare `DRIFT-MEDIAN` with seven different base-line methods, namely: Simple Averaging or Model Averaging (Wortsman et al., 2022; Choshen et al., 2022), Task Arithmetic (Ilharco et al., 2023), TIES (Yadav et al., 2023), DARE (Yu et al., 2024), Localize-and-Stitch (He et al., 2025a), Fisher merging (Matena & Raffel, 2022), and PCB-merging (DU et al., 2024). Hyperparameters detail are given in Appendix E.

**Models and Datasets** We conduct experiments on three different types of model architectures: 1) Llama family of models with various number of model parameters (Llama-3.1-8B, Llama-3.2-3B, Llama-2-7b), 2) GPT-2, and 3) CLIP-ViT Model. We evaluate `DRIFT-MEDIAN` on various diverse datasets. We refer the readers to Appendix D for more details.

**Evaluation Metric** When each task or domain includes multiple evaluation benchmarks of varying difficulty levels, a direct comparison of raw scores across tasks can be misleading. To obtain a fair comparison, we consider **Performance Retain Rate (PRR)** (He et al., 2025b) as evaluation

Table 3: Results on Llama-3.2-3B models; The results are reported in relative percentage w.r.t. the fine-tuned models.

| Method | Maths | Multilingual | Instruction | Coding | Safety | $\overline{\text{PRR}}$ |
|---|---|---|---|---|---|---|
| Model Averaging (excl. embed) | 63.08 | 101.69 | 56.81 | 88.11 | 57.81 | 73.50 |
| Model Averaging (incl. embed) | 64.83 | 101.79 | 51.74 | 86.36 | 56.46 | 72.24 |
| Task Arithmetic | **72.40** | 100.78 | 83.20 | 94.98 | 68.19 | 83.91 |
| TIES | 48.44 | **102.04** | 71.19 | 88.24 | 64.07 | 74.80 |
| DARE | 70.34 | 100.49 | 84.20 | **96.31** | 69.11 | 84.09 |
| Fisher Merging | 61.86 | 101.43 | 65.56 | 87.00 | 62.82 | 75.73 |
| Localize & Stitch | 68.59 | 101.63 | 56.81 | 86.36 | 49.00 | 72.48 |
| Ours | 65.77 | 99.96 | **100.49** | 93.80 | **72.39** | **86.48** |

Table 4: Results on Llama-2-7b models; The reported results for CMMLU, GSM8K and HumanEval are absolute values obtained from corresponding evaluation set.

| Method | CMMLU | GSM8K | HumanEval | $\overline{\text{PRR}}$ |
|---|---|---|---|---|
| Fine-tuned Models (DU et al., 2024) | 38.6 | 65.6 | 17.1 | - |
| Model Averaging | 35.6 | 47.8 | 8.5 | 71.60 |
| Task Arithmetic | 35.5 | 46.1 | 10.4 | 74.35 |
| TIES | 36.4 | 53.4 | 14 | 85.86 |
| PCB-Merging | 36.4 | 53.8 | 16.5 | 90.93 |
| Fine-tuned models (Ours) | 35.2 | 61.9 | 16.5 | - |
| Ours | 35.8 | 42.2 | 19.5 | 96.02 |

metric. PRR measures how much of the original performance of the task-specific fine-tuned model is retained by the merged model. Formally, for each task $t$, the PRR is defined as

$$\text{PRR}(t) = \frac{1}{N_t} \sum_{i=1}^{N_t} \frac{\text{Perf}(\theta^{\text{merge}}, D_{t,i})}{\text{Perf}(\theta^{(t)}, D_{t,i})} \times 100,$$

where $N_t$ is the number of evaluation benchmarks datasets for task $t$, $D_{t,i}$ denotes the $i$-th benchmark dataset for task $t$, $\text{Perf}(\theta, D)$ is the performance of model $\theta$ on dataset $D$, $\theta^{\text{merge}}$ denotes the merged model, and $\theta^{(t)}$ denotes the fine-tuned model on task $t$. This formulation normalizes the performance of the merged model against the best achievable performance for each benchmark (i.e., the fine-tuned baseline), and then averages across benchmarks within the task. By doing so, it avoids bias introduced by benchmarks of varying difficulty or scale, which would otherwise distort the results if raw scores were averaged directly. Thus, $\text{PRR}(t)$ provides a task-level measure of the degree to which the merged model retains the capabilities of the specialized fine-tuned models. Finally, we compute the mean PRR as $\overline{\text{PRR}} = 1/T \sum_{t=1}^{T} \text{PRR}(t)$ where $T$ is the total number of tasks. This overall score reflects the average retention of task-specific performance by the merged model, providing a single metric for multi-task evaluation.

### 4.2 RESULTS AND ANALYSIS

**Merging Fully fine-tuned GPT-2 Based Models:** For text classification, we adhere to the experimental setup of (Tang et al., 2024) for data and models. The setting considers a variety of text-classification tasks. We specifically consider 7 text-classification task (CoLA, MNLI, MRPC, QNLI, QQP, RTE, SST2) and report the experiments result in (Table 1). Individual cell except the last two columns of this table represents absolute accuracy. Last column represents the evaluation metric mean Performance Retain Rate. We also report mean accuracy of respective merging method to make a fair comparison with (Tang et al., 2024). Notably, DRIFT-MEDIAN outperforms all the baseline method and exceeds the best baseline method by 1.9 margin in mean PRR.

**Merging Fully fine-tuned Llama Based Models:** For generation tasks we consider Llama-3.1-8B, Llama-3.2-3B and Llama-2-7b model, we report experimental results in Table 2, Table 3 and Table 4 respectively. For Llama-3.1-8B and Llama-3.2-3B, we replicate the experiments of (He et al., 2025b) for respective data and models. In these two tables (Table 3 and Table 4), we consider

5 different task domains (Mathematics, Multilingual, Instruction, Coding, and Safety) with varying level of complexity. Each domain has multiple benchmarks. We first evaluate the fine-tuned model on the test set of corresponding benchmark. Detailed individual benchmark accuracies can be found out in Appendix F. Since each task includes multiple evaluation benchmarks of varying difficulty levels, so we represent the score in corresponding cell by its PRR. Finally, we compare the merging methods by mean PRR. Our model improves the baseline method by 3.76% and 2.39% for Llama-3.1-8B and Llama-3.2-3B, respectively. Note, in some cases multitasking models (here merged model) can exhibit slightly better performance than the individual fine-tuned model. Therefore, this may result in a PRR exceeding 100 in certain cases. In Table 4, we compare our method with the baseline method PCB-merging (DU et al., 2024). When we evaluate the fine-tuned models in respective task (sixth row), we get slightly different numbers than reported (first row) in the paper. So, for a fair comparison, first five rows in this table are copied from the paper (DU et al., 2024) and then we evaluate the merged methods in terms of mean PRR. In particular, our method outperforms PCB-merging by 5.09%.

**Merging Fully fine-tuned CLIP-ViT-B/32 Models** For image classification, we evaluate multi-task model merging across eight image classification datasets. Following (DU et al., 2024), we use the CLIP model (Radford et al., 2021) with ViT-B/32 as visual encoders. We have adopted same experimental configurations as Tang et al. (2024). We display our result in Table 5. While certain baseline methods occasionally outperform our approach on individual datasets, our method achieves superior overall performance compared with the leading baselines.

Table 5: Model Performance on Vision Tasks

| Method | SUN397 | CARS | RESISC45 | Eurosat | SVHN | GTSRB | MNIST | DTD | Average | $\overline{\text{PRR}}$ |
|---|---|---|---|---|---|---|---|---|---|---|
| Skyline(s) | 74.86 | 78.52 | 95.14 | 99.07 | 97.27 | 98.91 | 99.58 | 79.68 | 90.38 | |
| Task Arithmetic | 64.30 | 61.30 | 70.56 | 78.26 | 73.89 | 62.77 | 93.02 | 51.91 | 69.50 | 76.88 |
| TIES | 64.97 | 62.87 | **72.29** | 76.19 | 82.19 | 73.90 | 96.33 | 52.61 | 72.67 | 80.21 |
| Adamerging | 59.63 | 57.92 | 71.30 | **80.15** | 68.98 | 53.53 | 97.29 | 55.21 | 68.00 | 75.16 |
| PCB | 62.00 | 61.31 | 71.79 | 75.41 | 85.92 | **79.76** | **97.74** | 51.38 | 73.16 | 80.51 |
| Ours | **65.01** | **66.17** | 71.38 | 76.19 | **88.05** | 64.33 | 97.55 | **58.35** | 73.38 | **81.22** |

**Ablation of DRIFT-MEDIAN Components:** We systematically conduct ablation studies on each component of `DRIFT-MEDIAN` to evaluate their individual contributions. Beginning with the complete `DRIFT-MEDIAN` approach, we systematically eliminate or replace individual components and measure test set performance for the full model merging process. When we remove sign resolution step, we can clearly see the performance dip from 87.51% to 86.16%. In the Fisher weighted aggregation step of our method, if we consider mean instead of median we lose 2.29% of performance gain. If we eliminate the Top-$K$ selection component, performance drops drastically (2.41%). Further, we conducted an analysis on the first five GLUE tasks using GPT-2 (CoLA, MNLI, MRPC, QNLI, QQP). With a keep ratio of 60%, intra-model Top-K + sign election (as in TIES) leaves 5.89% of parameters with no surviving task update- i.e., no task contributes at those coordinates, forcing a fallback to the base model (parameter scarcity). In contrast, sign election + inter-model Top-K reduces this to 2.06%, meaning far fewer coordinates are left unused. This demonstrates that inter-model Top-K more closely

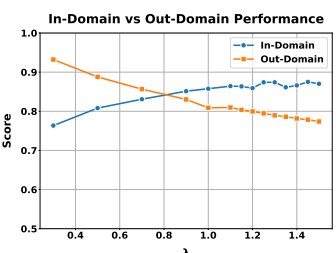

Figure 3: Sensitivity of $\lambda$ on in-domain and out-domain performance; Initially in-domain performance increases with increase in $\lambda$, reaching a saturation point then starts to decrease. The out-domain performance decrease with increase in $\lambda$.

matches the per-coordinate merging objective, reduces update scarcity, and more effectively utilizes the available task information. Finally, if we take out scaling step, we lose 1.74% performance. Table 6 demonstrates that each component of the method is essential for achieving optimal performance.

**Ablation on Hyperparameter $\lambda$:** For this ablation, we consider the same experimental setup and tasks given in the Table 2 for merging. We use a fixed keep ratio of 60% in this experiment. In Figure 3, we plot our model performance with respect to `DRIFT-MEDIAN` hyperparameter $\lambda$. It highlights the trade-off between in-domain specialization and out-domain generalization perfor-

Table 6: Ablation study on Llama-3.1-8B models; The results are in relative percentage w.r.t. the fine-tuned models.

| Method | Maths | Multilingual | Instruction | Coding | Safety | $\overline{\text{PRR}}$ |
|---|---|---|---|---|---|---|
| - With Top-$K$, Sign Resolution and Median | 85.19 | 89.44 | 71.24 | 94.45 | 101.47 | 87.51 |
| - w/o Sign Resolution | 86.66 | 91.40 | 63.09 | 93.22 | 101.18 | 86.16 |
| - Mean instead of Median | 88.82 | 91.88 | 57.68 | 93.61 | 97.19 | 85.22 |
| - w/o Top-$K$ | 88.25 | 96.10 | 68.54 | 88.08 | 92.40 | 85.10 |
| - w/o Scaling | 87.73 | 94.01 | 59.31 | 93.40 | 100.67 | 85.77 |

mance of our merged model with respect to $\lambda$. Here, $\lambda = 0$ signifies base model. Initially, with increase in $\lambda$ value, in-domain performance increases, reaching a saturation point then starts to decrease. The out-domain performance decrease with increase in $\lambda$. After certain value of $\lambda$, merging becomes unstable and performance of merged model drops significantly for in-domain and as well as out-domain. Our intended goal is to maximize performance on the known, in-domain tasks, and the out-domain results are reported primarily for completeness. If out-domain robustness were also an objective, one possible direction would be to adjust the trade-off parameter $\lambda$. Detailed descriptions of the specific task configurations for out-domain is given as General Domain paragraph in Appendix D.

**Hyperparameter Sensitivity**

To better understand the interaction between the Top-K and the scaling coefficient $\lambda$, we conduct a sensitivity study whose results are shown in Figure 4. The heatmap illustrates how different configurations of Top-K and $\lambda$ jointly affect mean PRR. As Top-K increases, more low-magnitude task deltas are included in the aggregation pool. These small updates, which lie very close to the base model, pull the merged parameter back toward the pretrained initialization. Consequently, configurations with higher Top-K values generally require a higher scaling coefficient $\lambda$ to counterbalance this pull and ensure that task-relevant updates maintain sufficient influence during aggregation. For this ablation, we consider the same experimental setup and tasks given in the Table 5 for merging.

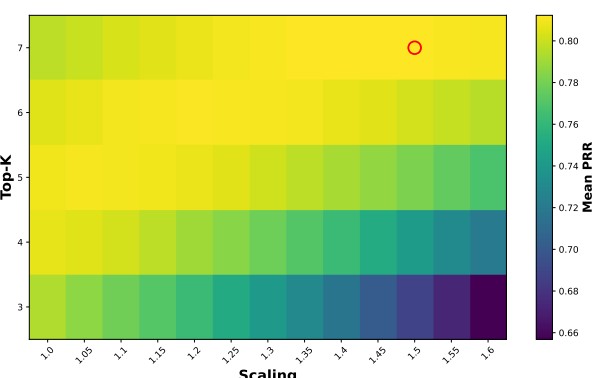

Figure 4: Sensitivity of the hyperparameters $K$ and $\lambda$ in `DRIFT-MEDIAN` on CLIP based tasks. The best performance is obtained on top-7 models with 1.5 as the scaling factor. The values near to the optimal hyperparameter have similar performance.

## 5 CONCLUSIONS

We propose `DRIFT-MEDIAN`, a task-aware model merging framework that combines task-vector sign resolution, coordinate-wise Top-$K$ selection, and Fisher-weighted median aggregation. By explicitly addressing sign disagreements and redundant-parameter interference, and incorporating parameter-sensitivity considerations, our method enables conflict-free parameter fusion while retaining task-specific knowledge. Experiments across mathematics, multilingual reasoning, coding, instruction following, and safety tasks demonstrate that `DRIFT-MEDIAN` consistently outperforms prior parameter-space merging methods such as TIES and Fisher merging. Potential directions for future work include the use of *dynamic* hyperparameters, where $\lambda$ and $\kappa$ adapt across models or even layers.

REPRODUCIBILITY STATEMENT

To ensure reproducibility, we provide a comprehensive description of our methodology together with derivation and a clear presentation of the proposed algorithm. Our work relies exclusively

on publicly available resources: all datasets, evaluation benchmarks, and pretrained model checkpoints used in this study are openly accessible to the research community. Upon acceptance, we will release the full source code, including training and evaluation scripts as well as detailed documentation, to facilitate independent verification and extension of our results. Furthermore, we specify all experimental details including hyper-parameters in Appendix D and Appendix E, ensuring that every component of our pipeline can be faithfully reproduced.

ETHICS STATEMENT

The datasets utilized in this research are openly accessible and linked to open license terms (CC-BY-2.0, CC-BY-4.0, MIT, Apache 2.0, Open Data Commons Attribution License, and Public Domain Dedication and License), which are suitable for research purposes. The datasets and models utilized in our experiments are explicitly cited and detailed in their corresponding sections. Models are downloaded from Huggingface. We acknowledge that we used LLMs in a limited capacity, solely for the purpose of grammatical refinement and sentences paraphrasing.

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

# A  ALGORITHM

---

**Algorithm 1** Fisher-Weighted Model Merging with Sign Resolution and Coordinate-wise Top-K Selection

---

**Input:** Base parameters $\boldsymbol{\theta}^{(0)} \in \mathbb{R}^N$, candidate models $\{\boldsymbol{\theta}^{(m)}\}_{m=1}^M$, Data $\{\mathcal{D}_m\}_{m=1}^M$, scaling factor $\lambda$, Keep ratio - $\kappa$
**Return:** Merged model $\boldsymbol{\theta}^{\text{merge}}$

1: Compute task vectors $\boldsymbol{\tau}^{(m)} \leftarrow \boldsymbol{\theta}^{(m)} - \boldsymbol{\theta}^{(0)}$
2: Compute directional sign $s_i \leftarrow \text{sign}\left(\sum_{m=1}^M \tau_i^{(m)}\right)$ for all $i$     ▷ Resolve sign step From TIES
3: **for** each $m = 1$ to $M$ **do**
4:     $\hat{\tau}_i^{(m)} = \begin{cases} \tau_i^{(m)}, & \text{if } \text{sign}(\tau_i^{(m)}) = s_i, \\ 0, & \text{otherwise.} \end{cases}$
5:     $\hat{\boldsymbol{\theta}}^{(m)} \leftarrow \boldsymbol{\theta}^{(0)} + \hat{\tau}^{(m)}$
6:     Compute Fisher $\boldsymbol{F}^{(m)}$ via empirical Fisher:
$$F^{(m)} = \mathbb{E}_{x \sim \mathcal{D}_m}\left[\left(\tfrac{\partial}{\partial \hat{\theta}^{(m)}} \log p(y \mid x; \hat{\boldsymbol{\theta}}^{(m)})\right)^2\right]$$
7: **end for**
8: $K = \lfloor \kappa \cdot M \rfloor$          ▷ Number of models to keep at each coordinate
9: **for** each coordinate $i$ **do**
10:     $\delta_i = \min\left(\text{Top-}K\left(\left\{\left|\hat{\tau}_i^{(m)}\right|\right\}_{m=1}^M\right)\right)$    ▷ Minimum deviation for consideration in merging
11:     $\mathcal{M}_i := \left\{m \in \{1, \dots, M\} : \left|\hat{\tau}_i^{(m)}\right| \geq \delta_i\right\}$
12:     $\mathcal{L}(\tau_i) = \sum_{m \in \mathcal{M}_i} F_i^{(m)} \cdot \left|\hat{\tau}_i^{(m)} - \tau_i\right|$
13:     $\tau_i^* = \arg\min_{\tau_i} \mathcal{L}(\tau_i)$          ▷ Closed form solution in Equation 4
14:     $\boldsymbol{\theta}_i^{\text{merge}} \leftarrow \boldsymbol{\theta}_i^{(0)} + \lambda \cdot \tau_i^*$
15: **end for**
16: Return final expert: $\boldsymbol{\theta}^{\text{merge}}$

---

# B  CLOSED-FORM SOLUTION FOR WEIGHTED $L_1$ LOSS: WEIGHTED MEDIAN

We are given the objective function:

$$\mathcal{L}(\tau_i) = \sum_{m \in \mathcal{M}_i} F_i^{(m)} \cdot \left|\hat{\tau}_i^{(m)} - \tau_i\right|,$$

where $\hat{\tau}_i^{(m)}$ are fixed scalar values and $F_i^{(m)} \geq 0$ are associated weights. Our goal is to find a value of $\tau_i$ that minimizes $\mathcal{L}(\tau_i)$.

PIECEWISE LINEARITY AND SUBGRADIENT

Each term $\left|\hat{\tau}_i^{(m)} - \tau_i\right|$ is convex and piecewise linear in $\tau_i$, with a non-differentiable point at $\tau_i = \hat{\tau}_i^{(m)}$. Thus, $\mathcal{L}(\tau_i)$ is convex and piecewise linear overall.

The subgradient with respect to $\tau_i$ is given by

$$\frac{\partial}{\partial \tau_i} F_i^{(m)} \cdot \left|\hat{\tau}_i^{(m)} - \tau_i\right| = \begin{cases} -F_i^{(m)} & \text{if } \tau_i < \hat{\tau}_i^{(m)}, \\ [-F_i^{(m)}, F_i^{(m)}] & \text{if } \tau_i = \hat{\tau}_i^{(m)}, \\ +F_i^{(m)} & \text{if } \tau_i > \hat{\tau}_i^{(m)}. \end{cases}$$

When $\tau_i$ does not coincide with any $\hat{\tau}_i^{(m)}$, the function is differentiable and its derivative is given by:

$$\frac{d\mathcal{L}}{d\tau_i} = - \sum_{\hat{\tau}_i^{(m)} < \tau_i} F_i^{(m)} + \sum_{\hat{\tau}_i^{(m)} > \tau_i} F_i^{(m)}.$$

Setting the derivative to zero yields:

$$- \sum_{\hat{\tau}_i^{(m)} < \tau_i} F_i^{(m)} + \sum_{\hat{\tau}_i^{(m)} > \tau_i} F_i^{(m)} = 0,$$

$$\Rightarrow \sum_{\hat{\tau}_i^{(m)} < \tau_i} F_i^{(m)} = \sum_{\hat{\tau}_i^{(m)} > \tau_i} F_i^{(m)} = \frac{1}{2} \sum_m F_i^{(m)}.$$

This condition defines the **weighted median**.

WHEN $\tau^* = \hat{\tau}^m$

For any $\tau$ (not necessarily median), let

$$A = \sum_{\hat{\tau}^{(m)} < \tau} F^{(m)}, \quad B = \sum_{\hat{\tau}^{(m)} = \tau} F^{(m)}, \quad C = \sum_{\hat{\tau}^{(m)} > \tau} F^{(m)},$$

so $A + B + C = T$.

Each term is convex, so

$$\partial \mathcal{L}(\tau) = \sum_{\hat{\tau}^{(m)} < \tau} (-F^{(m)}) + \sum_{\hat{\tau}^{(m)} = \tau} [-F^{(m)}, F^{(m)}] + \sum_{\hat{\tau}^{(m)} > \tau} F^{(m)} = [-A + C - B, \; -A + C + B].$$

Hence $0 \in \partial \mathcal{L}(\tau)$ iff

$$-A + C - B \le 0 \le -A + C + B \quad \Longleftrightarrow \quad |A - C| \le B.$$

SHOWING WEIGHTED MEDIAN SATISFIES THE CRITERIA

A point $\tau^*$ is called a *weighted median* if

$$A \le \frac{T}{2} \quad \text{and} \quad C \le \frac{T}{2}.$$

We show this implies $|A - C| \le B$:

1. $A \le \frac{T}{2}$ implies:
$$A \le \frac{A + B + C}{2}$$
$$\Rightarrow 2A \le A + B + C$$
$$\Rightarrow A \le B + C$$
$$\Rightarrow A - C \le B.$$

2. $C \le \frac{T}{2}$ implies:
$$C \le \frac{A + B + C}{2}$$
$$\Rightarrow 2C \le A + B + C$$
$$\Rightarrow C \le A + B$$
$$\Rightarrow C - A \le B.$$

3. Combining the two, we get:
$$|A - C| \le B.$$

Since $|A - C| \le B$ is exactly the condition for optimality, the weighted median minimizes $\mathcal{L}(\tau)$.

## C  IMPLEMENTATION DETAILS

A central design consideration in `DRIFT-MEDIAN` is the placement of different components in the merging pipeline. The calculation of the Fisher information matrix constitutes the primary computational bottleneck of our approach. To make the method practical, we decouple operations that require repeated hyperparameter tuning from those that do not. Specifically, the *Sign Resolution* step is performed prior to *Fisher Information Estimation*, since it does not involve any tunable parameters and can be fixed once for all runs. In contrast, the two hyperparameters of our method – keep ratio $\kappa$ for top-$K$ selection and scaling factor $\lambda$ —directly affect the aggregation and scaling stages. We, therefore, design the method such that these choices come *after* Fisher information estimation. This ensures that once the Fisher matrix is computed, it can be re-used efficiently for any combination of $\kappa$ and $\lambda$ without additional estimation overhead.

This design differs from prior work Lee et al. (2025), who perform scaling before Fisher merging and search over $\lambda$ across different models. While their approach is effective, it was applied primarily to much smaller models, where repeated Fisher estimation is less of a burden. In contrast, our design explicitly targets large-scale LLMs, where recomputing the Fisher matrix even a few times would be prohibitively expensive. By fixing Fisher estimation early and allowing hyperparameter flexibility afterward, `DRIFT-MEDIAN` achieves both scalability and adaptability.

## D  EVALUATION BENCHMARKS

We evaluate large language models (LLMs) across multiple domains using a diverse suite of benchmarks, each with carefully designed test sets. We evaluate `DRIFT-MEDIAN` on the following benchmark datasets: Minerva (Hendrycks et al., 2021), GSM8K (Cobbe et al., 2021), Harmbench (Mazeika et al., 2024), DAN (Shen et al., 2024), XSTest (Röttger et al., 2024), WildguardTest (Han et al., 2024), IFEval (Zhou et al., 2023), CMMLU (Li et al., 2024), 3 Multilingual Understanding tasks (Lai et al., 2023) (M_ARC, M_MMLU and M_HellaSwag), MBPP+ (Austin et al., 2021), HU-MANEVAL+ (Chen et al., 2021), and 7 GLUE (Wang et al., 2018; Warstadt et al., 2019) tasks (QQP, QNLI, RTE, CoLA, MRPC, MNLI and SST-2).

**Mathematics.** We consider two variants of the GSM8K (Cobbe et al., 2021) test set from the `lm-eval-harness`, namely GSM8K (5-shot) and GSM8K-CoT (8-shot). Since the test items (and gold answers) are identical, but model performance can vary depending on whether direct or chain-of-thought prompting is used, we report the best score across the two settings for each model. The GSM8K test set contains approximately 1.3k grade-school math word problems requiring multi-step reasoning and exact numeric answers. In addition, we include the Minerva Math (Lewkowycz et al., 2022) test set in a 4-shot setting, which consists of STEM-focused quantitative problems curated from the MATH benchmark (Hendrycks et al., 2021).

**Multilingual Understanding.** For cross-lingual evaluation, we employ translated test sets from three widely used benchmarks: M_ARC, M_MMLU, and M_HellaSwag (Lai et al., 2023). These test sets are direct multilingual extensions of the original English benchmarks, created via high-quality machine translation and covering multiple languages. We restrict evaluation to four representative languages: French (fr), German (de), Russian (ru), and Spanish (es) to assess reasoning and commonsense understanding across diverse linguistic settings. The test sets retain the multiple-choice structure of their English counterparts: M_ARC for science question answering, M_MMLU for multi-domain knowledge across 57 subjects, and M_HellaSwag for adversarial commonsense reasoning. To evaluate performance on Chinese on Llama2-7b models, we use the CMMLU (Li et al., 2024) benchmark .

**Instruction Following.** We evaluate using the IFEval (Zhou et al., 2023) test set, which contains 541 prompts covering 25 categories of verifiable instructions. Each prompt specifies explicit and automatically checkable constraints (e.g., output length, language, or formatting). In line with the original protocol, we report both *prompt-level strict accuracy*, which requires that all constraints be satisfied exactly, and *prompt-level loose accuracy*, which allows multiple post-processing transformations of the model output and considers a response correct if any transformed version meets all specified criteria.

**Code Generation.** We adopt the HumanEval+ (Chen et al., 2021) and MBPP+ (Austin et al., 2021) test sets from the EvalPlus framework, which augment the original HumanEval and MBPP problems with substantially more hidden test cases (approximately $80\times$ more for HumanEval and $35\times$ more for MBPP). We report the *Pass@1* metric across these test sets. For consistency with prior work on PCB merging, we evaluate **Llama-2-7b** using the original HumanEval test set of 164 handwritten programming tasks, ensuring comparability with published results.

**Safety and Robustness.** To assess safety, we employ several adversarial and red-teaming test sets. The WildGuardTest (Han et al., 2024) set contains $\sim$5k human-annotated examples from WildGuardMix, labeled across 13 harm categories and evaluated for prompt harmfulness, response harmfulness, and refusal detection. The HarmBench (Mazeika et al., 2024) test suite provides a standardized set of adversarial prompts for automated red-teaming, enabling direct measurement of attack success rates and robust refusal behavior. In addition, we include adversarial jailbreak prompts from the DAN (Do Anything Now) (Shen et al., 2024) family, which are widely used to probe model vulnerabilities in controlled settings. Finally, we use the XSTest (Röttger et al., 2024) benchmark, which comprises 250 safe prompts and 200 unsafe prompts designed to evaluate both over-refusal (failing to answer benign queries) and under-refusal (incorrectly answering harmful queries).

**Natural Language Understanding (GLUE).** For GPT2, we evaluate models on the GLUE benchmark (Wang et al., 2018). Specifically, we include the following tasks: CoLA (linguistic acceptability), MNLI (multi-genre natural language inference), MRPC (paraphrase detection), QNLI (Question Natural Language Inference), QQP (Quora question pairs), RTE (textual entailment), and SST-2 (Stanford Sentiment Treebank). We use the fine-tuned checkpoints from Fusion-Bench (Tang et al., 2024) library.

**Vision Datasets** Following PCB merging (DU et al., 2024), we consider multi-task model merging across eight image classification datasets. SUN397 (Xiao et al., 2016) comprises of 397 classes of scene images. Stanford Cars (Krause et al., 2013) is car classification dataset consisting of 196 car classes. RESISC45 (Cheng et al., 2017) consist of 45 classes of remote sensing image scenes. EuroSAT (Helber et al., 2019) includes 10 classes of geo-referenced satellite images. SVHN (Netzer et al., 2011) contains 10 classes of real-world digital classification images. GTSRB (Stallkamp et al., 2011) features 43 classes of traffic signs. MNIST (LeCun, 1998) consists of grayscale handwritten digits across 10 classes. Finally, DTD (Cimpoi et al., 2014)is a texture classification dataset with 47 classes.

**General Domain** To assess broader reasoning and domain generalization, we include several widely used benchmarks: CoQA (Reddy et al., 2019), MMLU (Hendrycks et al., 2021), PubMedQA (Jin et al., 2019), SQuADv2 (Rajpurkar et al., 2018), and TriviaQA (Joshi et al., 2017). CoQA measures conversational question answering with context-dependent reasoning, while MMLU evaluates multi-domain expert knowledge across 57 subjects. PubMedQA focuses on biomedical question answering, enabling evaluation in a specialized scientific domain. SQuADv2 extends extractive QA with unanswerable questions, testing robustness in distinguishing relevant from irrelevant contexts. TriviaQA probes open-domain QA with a mix of factoid and reasoning-intensive queries. Together, these benchmarks capture general-purpose reasoning, knowledge retrieval, and robustness across domains.

## E  HYPERPARAMETERS AND COMPUTATION REQUIREMENTS

To identify suitable hyperparameter configurations for our proposed DRIFT-MEDIAN framework, we initially conducted exploratory searches on GPT-2, owing to its relatively small size and faster training and inference cycles. In this setting, we varied the Top-$K$ parameter from Top-1 through Top-7, and also evaluated the *keep-above-mean* and *keep-above-median* strategies. The sweep over $\lambda$ was deliberately non-uniform: we first sampled random values across the full interval $[0.3, 3.0]$ and observed that the strongest performance consistently occurred when $\lambda$ lay in the narrower band of approximately 1.1–1.5. We then performed a denser search within this region using 0.05 increments.

From these experiments, we found that the best configuration on GPT-2 corresponded not to a fixed Top-$K$ selection, but rather to a thresholding strategy, where all coordinates above the mean are retained, combined with a scaling value of $\lambda = 1.35$. For larger LLMs, we subsequently searched in the neighborhood of these optimal GPT-2 values. For Llama-3.1-8B, the best performance was achieved with Top-3 and $\lambda = 1.45$, while for Llama-3.2-3B, the best performance was obtained with Top-3 and $\lambda = 1.30$. For Llama-2-7b, the best configuration was Top-2 with $\lambda = 1.20$. It is important to note that aggressive hyperparameter search is not required. DRIFT-MEDIAN maintains good performance even when $\lambda$ and $K$ are set near the optimal values, demonstrating that the method is robust and not overly sensitive to hyperparameter choices. For baseline comparisons, we adopt the hyperparameter settings recommended by MergeBench He et al. (2025b) for Llama-3.2-3B and Llama-3.1-8B models. Specifically, for TIES we use Top-$K = 0.3$ and a scaling factor of $\lambda = 0.4$, for DARE we set the sparsity to $0.9$ with $\lambda = 0.4$, and for L&S we use a sparsity of $0.1$. Since we were unable to reproduce the with-data version of L&S due to hardware constraints, we report results for the dataless variant.

The most expensive step in our pipeline is the computation of Fisher information matrices. For GPT-2, we use 256 examples, while for larger LLMs we use 1000 examples, which is the maximum available from the validation data. On a single NVIDIA A100 80GB GPU, Fisher estimation for the 8B model takes approximately one hour per domain, which results in about five hours of computation for five tasks. Crucially, this cost is incurred only once per task since Fisher estimation is independent of hyperparameters. Once the Fisher values are computed, they can be reused for all subsequent searches over $\lambda$ and $K$, significantly reducing the overhead of iterative experimentation. Moreover, Fisher computation is naturally parallelizable: different tasks' Fisher information can be computed simultaneously, and multi-GPU setups can further accelerate the process. While we did not explore these parallelization strategies in this work, they represent a clear avenue for further speedup in large-scale deployments.

# F  DETAILED RESULTS

Table 7: Model Performance: Mathematics, Multilingual, and Instruction Following Tasks on Llama-3.1-8B

| Model Name | GSM8K | Minerva | M_MMLU | M_ARC | M_Hellaswag | IFEval (strict) | IFEval (loose) |
|---|---|---|---|---|---|---|---|
| Base Model | 55.72 | 18.00 | 53.28 | 44.75 | 63.08 | 18.70 | 20.10 |
| Skyline(s) | 70.43 | 33.22 | 54.02 | 47.62 | 65.35 | 57.70 | 63.60 |
| Averaging (only layers.) | 71.27 | 25.40 | 52.68 | 43.51 | 65.90 | 24.80 | 28.80 |
| Averaging (All) | 71.49 | 25.76 | 52.70 | 43.81 | 65.94 | 23.80 | 27.40 |
| Task Arithmetic | 75.66 | 28.02 | 49.08 | 41.22 | 64.03 | 30.90 | 37.20 |
| TIES | 76.72 | 28.22 | 52.26 | 43.02 | 65.83 | 28.80 | 33.80 |
| DARE | 75.06 | 27.02 | 47.32 | 40.86 | 62.81 | 30.10 | 35.90 |
| Fisher merging | 67.32 | 22.60 | 52.67 | 43.62 | 65.66 | 34.60 | 39.70 |
| L&S | 76.12 | 28.60 | 53.49 | 44.04 | 65.01 | 24.20 | 30.90 |
| Ours | 64.67 | 23.28 | 47.67 | 40.22 | 62.48 | 40.30 | 46.20 |

Table 8: Model Performance: Coding and Safety Tasks on Llama-3.1-8B

| Model Name | HumanEval+ | MBPP+ | WildguardTest | Harmbench | DAN | XSTest |
|---|---|---|---|---|---|---|
| Base Model | 31.70 | 51.30 | 42.19 | 24.69 | 29.67 | 34.22 |
| Skyline(s) | 57.30 | 54.80 | 78.37 | 81.56 | 71.33 | 69.11 |
| Averaging (only layers.) | 46.30 | 54.00 | 58.88 | 44.06 | 59.67 | 60.22 |
| Averaging (All) | 44.50 | 55.60 | 60.35 | 41.87 | 58.00 | 65.56 |
| Task Arithmetic | 49.40 | 52.40 | 59.81 | 48.75 | 58.33 | 69.56 |
| TIES | 46.30 | 54.80 | 61.82 | 56.25 | 67.67 | 64.22 |
| DARE | 45.10 | 52.60 | 60.35 | 50.73 | 51.33 | 68.44 |
| Fisher merging | 44.50 | 53.20 | 68.89 | 57.50 | 77.00 | 60.45 |
| L&S | 42.70 | 53.40 | 51.27 | 40.31 | 42.00 | 49.33 |
| Ours | 52.40 | 53.40 | 75.83 | 75.62 | 78.67 | 73.33 |

Table 9: Model Performance: Mathematics, Multilingual, and Instruction Following Tasks on Llama-3.2-3B

| Model Name | GSM8K | Minerva | M_MMLU | M_ARC | M_Hellaswag | IFEval (strict) | IFEval (loose) |
|---|---|---|---|---|---|---|---|
| Base Model | 28.66 | 8.04 | 45.32 | 36.75 | 55.45 | 20.0 | 21.6 |
| Skyline(s) | 55.50 | 23.68 | 44.12 | 39.19 | 56.45 | 37.9 | 43.8 |
| Averaging (only layers.) | 40.86 | 12.44 | 45.83 | 38.94 | 57.49 | 21.6 | 24.8 |
| Averaging (All) | 41.77 | 12.88 | 45.87 | 39.00 | 57.52 | 19.4 | 22.9 |
| Task Arithmetic | 44.45 | 15.30 | 44.27 | 39.22 | 57.54 | 32.0 | 35.9 |
| TIES | 31.46 | 9.52 | 45.43 | 39.66 | 57.55 | 27.4 | 30.7 |
| DARE | 43.44 | 14.78 | 43.89 | 39.34 | 57.36 | 32.5 | 36.2 |
| Fisher merging | 40.26 | 12.12 | 45.98 | 38.64 | 57.28 | 24.6 | 29.0 |
| L&S | 42.76 | 14.24 | 45.20 | 39.73 | 57.05 | 21.6 | 24.8 |
| Ours | 40.33 | 13.94 | 44.33 | 38.51 | 57.10 | 38.1 | 44.0 |

Table 10: Model Performance: Coding and Safety Tasks on Llama-3.2-3B

| Model Name | HumanEval+ | MBPP+ | WildguardTest | Harmbench | DAN | XSTest |
|---|---|---|---|---|---|---|
| Base Model | 25.0 | 39.4 | 26.70 | 26.25 | 29.33 | 28.67 |
| Skyline(s) | 38.4 | 45.8 | 85.71 | 88.75 | 90.67 | 38.67 |
| Averaging (only layers.) | 31.7 | 42.9 | 37.38 | 35.62 | 37.33 | 41.11 |
| Averaging (All) | 31.7 | 41.3 | 37.92 | 34.37 | 37.33 | 39.33 |
| Task Arithmetic | 34.8 | 45.5 | 51.67 | 39.37 | 34.67 | 50.22 |
| TIES | 32.3 | 42.3 | 47.00 | 40.94 | 46.00 | 40.44 |
| DARE | 35.4 | 46.0 | 49.00 | 37.19 | 33.67 | 54.22 |
| Fisher merging | 31.1 | 42.6 | 48.06 | 39.06 | 48.00 | 38.00 |
| L&S | 31.7 | 41.3 | 30.71 | 27.81 | 26.67 | 38.44 |
| Ours | 35.4 | 43.7 | 56.74 | 37.81 | 46.67 | 50.00 |

# G    SCALABILITY OF `DRIFT-MEDIAN`

The proposed method should be applicable to larger sized models. Computing the diagonal Fisher information matrix requires only forward–backward passes over a small validation set and does not demand more resources than a brief fine-tuning run. In practice, we did not conduct such experiments due to two limitations: (i) a lack of computational resources to load the models, and (ii) the absence of large, publicly available fine-tuned checkpoints built on the same base model, which makes controlled comparisons difficult. Further, understanding how performance behaves as the number of merged tasks increases is an important question for evaluating the robustness of any parameter-space merging method. However, this behavior does not depend solely on the task count; it is shaped by several interacting factors, including task similarity, parameter-space overlap, sparsity patterns of the task vectors, and the curvature or variance induced by each task. As noted in Wang et al. (2025), the effective parameter space can saturate as more experts are merged due to Gaussian Width concavity and redundancy constraints. This implies that performance may plateau or even degrade when the additional tasks introduce conflicting or redundant update directions, independent of their absolute number. A faithful study of this phenomenon requires a carefully controlled setting that isolates these effects, which is beyond the scope of the present work.

# H    CORRELATION WITH TASK-VECTOR MAGNITUDE

To better understand whether the degree of parameter deviation in each task influences the final merged performance, we compare the difference between our method and the Skyline(s) baseline with the mean absolute magnitude of the corresponding task vectors. The performance difference is computed as Ours – Skyline, capturing how much accuracy is lost relative to the ideal single-task fine-tuned models. The mean task-vector magnitude reflects the average absolute parameter shift introduced by each task relative to the pretrained backbone. Figure 5 summarizes these two quantities across all eight datasets. The performance differences span a much wider range, from modest degradation (e.g., MNIST) to substantial drops (e.g., GTSRB and RESISC). Importantly, datasets with relatively higher task-vector magnitudes do not consistently exhibit lower deviations from Skyline performance, and tasks with lower magnitudes do not show systematically higher gaps. Overall, this analysis indicates that the magnitude of the task-vector updates is not strongly

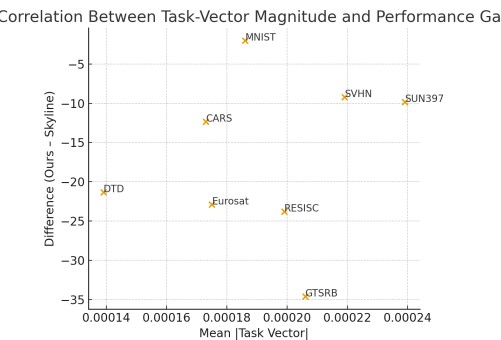

Figure 5: Relationship between task-vector magnitude and performance degradation in the merged model. Despite small variations in the average parameter shift across tasks, no meaningful correlation emerges, indicating that other factors such as dataset difficulty or task conflicts may dominate the observed performance differences.

correlated with the degradation observed when merging models. The variation in performance across datasets is therefore more likely driven by dataset-specific difficulty or inherent conflicts between task objectives rather than by simple differences in the size of the underlying task vectors.

# I   DOMAIN SENSITIVITY OF `DRIFT-MEDIAN`

Table 11: Model Performance on Different Domain Data for Fisher Estimation

| Validation Data | SUN397 | CARS | RESISC45 | Eurosat | SVHN | GTSRB | MNIST | DTD | Average | $\overline{\text{PRR}}$ |
|---|---|---|---|---|---|---|---|---|---|---|
| Unchanged | 65.01 | 66.17 | 71.38 | 76.19 | 88.05 | 64.33 | 97.55 | 58.35 | 73.38 | 81.22 |
| MNIST → KMNIST | 65.17 | 66.45 | 71.11 | 75.96 | 86.83 | 64.31 | 98.06 | 58.09 | 73.25 | 81.09 |
| MNIST → KMNIST & SVHN → MNIST | 65.33 | 66.92 | 71.98 | 77.19 | 77.43 | 64.65 | 97.36 | 58.35 | 72.40 | 80.25 |

To further analyze the robustness of `DRIFT-MEDIAN`, we additionally study the effect of domain mismatch in the Fisher estimation stage. Specifically, we replace the MNIST validation data with KMNIST, a visually distinct digit-recognition dataset where the characters correspond to Japanese cursive hiragana Clanuwat et al. (2018). Despite the significant visual shift from English numerals, performance across domains remains relatively stable, demonstrating that `DRIFT-MEDIAN` tolerates moderate domain shifts. We chose KMNIST because both MNIST and KMNIST share the same label space (0 to 9).

In the last row of Table 11, we perform a more extreme modification by replacing SVHN (Street View House Numbers), which contains real-world RGB street-number images, with MNIST grayscale digits. In this case, we observe a substantial performance drop on SVHN, while the other domains remain consistent. This behavior is expected because SVHN contains cluttered and noisy backgrounds and RGB images whereas MNIST contains grayscale images. Together, these results show that `DRIFT-MEDIAN` is robust to moderate domain shifts in the Fisher estimation data but can degrade when the substitute domain differs too drastically from the target distribution. Importantly, in Table 2 and Table 3, we use validation data that do not exactly match the downstream evaluation tasks. For example, we use multilingual instruction-following data to compute Fisher information, while the evaluation is performed on tasks such as ARC, HellaSwag, and MMLU. Similarly, for other LLM benchmarks, including MBPP, Humaneval, and GSM8K, there are no official validation sets available. In all the cases, we rely on datasets whose topical focus may be broadly related, but whose style and distributions differ substantially from the downstream tasks. Even though these datasets differ from the evaluation sets, `DRIFT-MEDIAN` maintains strong performance for all models.

In conclusion, `DRIFT-MEDIAN` is generally resilient to reasonable domain mismatch and can operate effectively even when the Fisher estimation data and evaluation data come from different distributions. However, extremely mismatched domains such as replacing SVHN with MNIST can negatively impact performance.

## J    USE OF LARGE LANGUAGE MODELS

In adherence to ICLR 2026 policy, we disclose our use of Large Language Models (LLMs) during the preparation of this manuscript. ChatGPT (OpenAI et al., 2024) was utilized in a limited capacity as a general-purpose writing assistant for grammatical refinement and sentence paraphrasing. The core research ideas, experimental design, results, and their interpretation were conceived and formulated entirely by the authors. The LLM's role was strictly limited to language refinement and did not contribute to the scientific ideation or analysis presented in this work. The authors have reviewed all content and take full responsibility for the final manuscript.

