# OpenReview forum: "Task-Aware Model Merging via Fisher-Weighted Median"
_ICLR.cc/2026/Conference — ICLR 2026 Conference Withdrawn Submission_

### Official Review · Reviewer_r1B3 · 2025-10-22

**Soundness:** 2
**Presentation:** 3
**Contribution:** 2
**Rating:** 2
**Confidence:** 3

**Summary:**

The paper proposes DRIFT-MEDIAN, a new parameter-space model merging framework for combining LLMs across different tasks without retraining. The authors introduce a Fisher-weighted median aggregation mechanism for merging parameters robustly. Experiments are conducted on GPT-2, Llama-3.1-8B, Llama-3.2-3B, and Llama-2-7B, covering diverse domains such as math reasoning, coding, multilingual understanding, instruction following, safety, and GLUE benchmarks.

**Strengths:**

The authors evaluate across a wide range of model architectures and task types across language models.

This paper is well-written and easy to follow.

**Weaknesses:**

Although the combination of Fisher weighting and interference mitigation is new, each individual component (sign pruning, Top-K selection, Fisher weighting) already exists. The paper’s novelty is limited.

The method’s theoretical advantages are not quantitatively demonstrated. For instance, it would help to show why the Fisher-weighted median is superior to Fisher-weighted mean beyond robustness claims.

Fisher Information estimation for large models (8B parameters) is computationally expensive—reported as “~1 hour per domain on A100” (Appendix E). The practicality for extremely large-scale merging such as 14B or 32B or real-world continual learning scenarios remains questionable.

Although the proposed method is applied to language tasks. It is recommended to provide results on vision-language models, such as merging eight CLIP models over 8 classification datasets.

While λ is analyzed, the keep ratio κ (which controls sparsity) is not systematically explored. Its effect on stability and generalization should be reported.

**Questions:**

See weakness. I think the major one is the limited novelty.

---

> ### Author Response · Authors · 2025-11-21
>
> ## Weakness - 1 and Weakness - 2
> > Although the combination of Fisher weighting and interference mitigation is new, each individual component (sign pruning, Top-K selection, Fisher weighting) already exists. The paper’s novelty is limited.
>
>
> > The method’s theoretical advantages are not quantitatively demonstrated. For instance, it would help to show why the Fisher-weighted median is superior to Fisher-weighted mean beyond robustness claims.
>
> A key challenge in parameter-space merging is that both extremes: staying too close to the base model or drifting too far from it are harmful. If the merged parameters remain too close to the base, the model fails to capture the meaningful task-specific knowledge encoded in the fine-tuned task vectors, making it under-specialized model. Conversely, if the merge drifts too far toward any individual task vector, the merged model can be pulled away from the pretrained manifold. This often leads to degradation on unrelated tasks. A merging method therefore should maintain a balance between them.
> This is where Top-K and the Fisher-weighted median operate in a complementary way. Coordinate-wise Top-K selects the most meaningful task updates at each coordinate, preventing the merged parameter from being overwhelmed by a large number of small deltas. However, Top-K alone cannot prevent extreme but low-Fisher updates from pulling the merged parameter too far from the base.
> Fisher-weighted median ensures that the merged parameter will move toward an extreme update only when that direction carries more than half of the total Fisher mass among the Top-K survivors. In other words, Top-K identifies the competitive candidate updates, while the median ensures that only those that are genuinely important (according to Fisher) can dominate the final merged value. This prevents excessive drift away from the base while still enabling meaningful movement toward task-relevant directions. The complementary nature of these two components is also evident in Table 6. Replacing the weighted median with a weighted mean yields slightly better scores on a few tasks but causes large degradations on others, resulting in a worse overall PRR.
>
> Another key contribution of our work is the closed-form solution based on the Fisher-weighted median, which ensures that parameter contributions reflect both sensitivity and relevance, leading to more robust merging. We further show that the closed-form solution to the weighted L1 objective can be expressed (Appendix B) directly in terms of the Fisher-weighted median.
>
> ## Weakness - 3
> > Fisher Information estimation for large models (8B parameters) is computationally expensive—reported as “~1 hour per domain on A100” (Appendix E). The practicality for extremely large-scale merging such as 14B or 32B or real-world continual learning scenarios remains questionable.
>
> Thank you for raising this concern. We agree that Fisher computation deserves clearer discussion. In practice, computing the diagonal Fisher matrix is not expensive than ordinary training steps, since it uses exactly the same operations: forward pass, loss computation, and backward pass but without the optimizer update. Thus, if an environment supports fine-tuning of large model, it can also support Fisher estimation, with cost driven mainly by lower batch size rather than by Fisher-specific overhead. We also acknowledge that continual-learning settings may require repeated Fisher computations and therefore incur higher cost; however, prior continual-learning work (Schwarz et al. 2018) demonstrates that parameter importance can be approximated using gradients already produced during training, offering a practical, low-overhead alternative to explicit Fisher passes. We will clarify these points in the revised manuscript.
>
> ## Weakness - 4
> > Although the proposed method is applied to language tasks. It is recommended to provide results on vision-language models, such as merging eight CLIP models over 8 classification datasets.
>
> We thank the reviewer for this valuable suggestion. In response, we have added results on eight CLIP models across eight classification datasets, and included these results in Table 5 of the revised manuscript. This additional experiment demonstrates that our method also transfers effectively beyond pure language tasks.
> ## Weakness - 5
> > While λ is analyzed, the keep ratio κ (which controls sparsity) is not systematically explored. Its effect on stability and generalization should be reported.
>
> In the revised manuscript, we now include a systematic analysis of the keep-ratio  in Figure 4. As shown plot, the method remains stable across a wide range of κ values near the optimal setting. This indicates that DRIFT-MEDIAN is robust to changes in sparsity, given that scaling is adjusted accordingly. We appreciate the reviewer’s suggestion to report this.

---

> ### Author Response · Authors · 2025-11-21
>
> **References**
> - Jonathan Schwarz, Wojciech Czarnecki, Jelena Luketina, Agnieszka Grabska-Barwinska, Yee Whye Teh, Razvan Pascanu, Raia Hadsell Proceedings of the 35th International Conference on Machine Learning, PMLR 80:4528-4537, 2018.

---

> > ### Comment · Reviewer_r1B3 · 2025-11-27
> >
> > Thank you for your detailed response and for taking the time to clarify the intuition behind your method. I appreciate the explanation.
> >
> > However, my main concerns still feel unresolved. In particular:
> >
> > The novelty of the proposed approach does not yet appear clearly distinguishable from existing work.
> >
> > The use of the Fisher matrix seems to introduce substantial computational overhead. I’m not fully convinced that using finetuning to estimate Fisher. In most cases, we directly obtain finetuned checkpoints, such as downloading from Hugging Face.

---

### Official Review · Reviewer_8UxF · 2025-10-26

**Soundness:** 2
**Presentation:** 3
**Contribution:** 2
**Rating:** 4
**Confidence:** 5

**Summary:**

This paper proposes DRIFT-MEDIAN, a framework for combining multiple fine-tuned LLMs into a single multitask model. The method integrates key advances from both parameter interference reduction and sensitivity-aware merging. To achieve this, it employs a multi-stage process that first mitigates interference using sign resolution and coordinate-wise Top-K filtering. The refined parameters are then aggregated via a Fisher-weighted median. Comprehensive experiments on GPT-2 and Llama variants across a diverse range of benchmarks demonstrate that DRIFT-MEDIAN consistently outperforms existing model merging baselines.

**Strengths:**

1. The paper is clearly written and well-organized. The architectural pipeline illustrated in Figure 1 provides a clear step-by-step overview of the methodology, making a complex multi-stage process easy to understand.
2. The proposed method is thoroughly evaluated across a diverse set of LLM architectures (GPT-2, Llama-2, and two Llama-3 variants) and a wide range of tasks, including mathematics, coding, multilingual reasoning, and safety. This comprehensive evaluation increases confidence in the generalizability and robustness of the results.
3. The authors provide a detailed description of the experimental setup, including baseline methods, datasets, and hyperparameter settings. This significantly aids in the reproducibility of the work.

**Weaknesses:**

1. **Limited Novelty**: The main weakness is the incremental nature of the contribution. The proposed framework is largely a synthesis of well-established techniques:

  - The sign resolution step is directly adopted from TIES-merging [1].
  - The use of Fisher information to weight parameter importance is the core idea of Fisher-merging [2].
  - Top-K filtering is a key component of TIES-merging [1] and SCE-merging [3], which the authors adapt to a coordinate-wise application.
  - The final scaling of the merged task vector is a standard technique used in Task Arithmetic [4] and its successors.

While the combination is effective, the paper primarily rearranges and refines existing building blocks rather than introducing a new, foundational concept.

2. **Potential Overclaiming of Contribution**: The abstract (lines 22-25) and introduction state that interference-reduction and sensitivity-based approaches have remained "largely disconnected" and that this work aims to "bridge the gap". This claim may be too strong. Recent works such as SCE-merging [3] and PCB-merging [5], also implicitly consider both interference and parameter importance. The authors should more carefully contextualize their work with respect to these methods.

**References**:

[1] Ties-merging: Resolving interference when merging models. (Yadav, et al., NeurIPS 2023)

[2] Merging models with fisher-weighted averaging. (Matena, et al., NeurIPS 2022)

[3] Fusechat: Knowledge fusion of chat models. (Wan, et al., Arxiv 2024)

[4] Editing models with task arithmetic. (Ilharco G, et al., ICLR 2023)

[5] Parameter competition balancing for model merging. (Du, et al., NeurIPS 2024)

**Questions:**

1. Could you please elaborate on what you see as the primary conceptual contribution of this work beyond the successful integration of existing methods? What is the core, fundamental problem in model merging that DRIFT-MEDIAN solves in a way that prior art has not?
2. The visualization in Figure 2 contrasts the "model-wise" Top-K selection of TIES with your "coordinate-wise" approach. This "row-wise" vs. "column-wise" distinction, however, seems contingent on the arbitrary representation of parameters and models as matrix axes. Could you provide a more fundamental rationale for why applying Top-K filtering independently at *each parameter coordinate* is superior? A more detailed justification or supplementary analysis would strengthen this design choice.
3. Have you investigated the interplay between the keep-ratio $K$ and scaling factor $\lambda$? For instance, does a more aggressive pruning (smaller $K$) necessitate a larger scaling factor  $\lambda$ to compensate? A 2D sensitivity plot for these two parameters would be insightful.
4. While the method shows strong average performance, a qualitative analysis of its limitations would be valuable. For instance, the paper reports performance degradation on certain tasks (e.g., Multilingual in Table 2, Maths in Table 3). Could the authors provide an analysis of these failure cases? What might such an analysis reveal about the method's underlying assumptions or potential weaknesses?
5. The paper provides limited detail regarding the computation of the Fisher information matrix, which is a critical component. Could you please clarify the following:

- **Scalability:** How does the computational cost of this step scale with model size? Is it linear, or does it pose a significant bottleneck for even larger models?

- **Sample Selection:** What criteria were used to select the samples for this computation? What is the composition of the "validation data" mentioned in line 978, how was it constructed, and is it fully disjoint from the test sets used for final evaluation?

- **Domain Sensitivity:** Have you analyzed how the domain coverage and diversity of the dataset used for Fisher estimation affect the final merging performance? For example, would using a more general-purpose dataset instead of a task-specific one alter the results?

6. I am willing to raise my score if the authors can satisfactorily address the weaknesses and questions I have raised.

---

> ### Author Response · Authors · 2025-11-21
>
> ## Weakness - 1 and Question - 1
> > Limited Novelty: The main weakness is the incremental nature of the contribution. The proposed framework is largely a synthesis of well-established techniques: The sign resolution step is directly adopted from TIES-merging. The use of Fisher information to weight parameter importance is the core idea of Fisher-merging. Top-K filtering is a key component of TIES-merging and SCE-merging, which the authors adapt to a coordinate-wise application. The final scaling of the merged task vector is a standard technique used in Task Arithmetic and its successors. While the combination is effective, the paper primarily rearranges and refines existing building blocks rather than introducing a new, foundational concept.
> > Could you please elaborate on what you see as the primary conceptual contribution of this work beyond the successful integration of existing methods? What is the core, fundamental problem in model merging that DRIFT-MEDIAN solves in a way that prior art has not?
>
> We thank the reviewer for the comment. While our method indeed builds on components from TIES-Merging and Fisher-Merging, the core conceptual contribution is not the components themselves but the new way they are combined to solve a problem: how to obtain, at each parameter coordinate, a merged value that is simultaneously (i) free of destructive interference, (ii) aligned with parameter sensitivity, and (iii) protected against parameter crowding/scarcity across tasks. Existing methods address these aspects only in isolation—TIES handles sign conflicts and pruning but ignores sensitivity, while Fisher-Merging incorporates sensitivity but cannot control extreme task deltas or cross-task competition. DRIFT-MEDIAN introduces a new Fisher-weighted consensus formulation together with coordinate-wise competition control, ensuring that updates move away from the base model only when Fisher-validated consensus exists among tasks. This ensures a more balanced performance among tasks.
> ## Weakness - 2
> > Potential Overclaiming of Contribution: The abstract (lines 22-25) and introduction state that interference-reduction and sensitivity-based approaches have remained "largely disconnected" and that this work aims to "bridge the gap". This claim may be too strong. Recent works such as SCE-merging and PCB-merging, also implicitly consider both interference and parameter importance. The authors should more carefully contextualize their work with respect to these methods.
>
> While recent approaches such as SCE-merging and PCB-merging touch on aspects related to interference and parameter importance, they differ fundamentally from our sensitivity-based formulation. SCE uses magnitude-based heuristics (variance filtering and squared update norms) and sign-consistency rules; these capture how much parameters change, but not how sensitive model performance is to perturbations in those parameters. PCB, balances inter-model and intra-model task vector competitions, but it is not dependent upon task-specific parameter sensitivity. We have added a short discussion on SCE-merging and PCB merging in the related works section.

---

> ### Author Response · Authors · 2025-11-21
>
> ## Question - 2
> > The visualization in Figure 2 contrasts the "model-wise" Top-K selection of TIES with your "coordinate-wise" approach. This "row-wise" vs. "column-wise" distinction, however, seems contingent on the arbitrary representation of parameters and models as matrix axes. Could you provide a more fundamental rationale for why applying Top-K filtering independently at each parameter coordinate is superior? A more detailed justification or supplementary analysis would strengthen this design choice.
>
> Thank you for the opportunity to clarify this. The distinction in Figure 2 is not about matrix orientation but about where sparsification is applied: intra-model vs. inter-model. TIES applies Top-K within each task vector, selecting the largest magnitude coordinates independently for each model. Our method applies Top-K across models at each parameter index, which aligns with the actual merging objective: at every coordinate of the merged model we must produce one final value, so the relevant competition is between tasks at that coordinate, not between coordinates within a task.
>
> With intra-model Top-K, several tasks may propose large updates at the same coordinate, creating multiple competing candidates to reconcile while many other coordinates receive no task-specific update at all, forcing a fallback to the base (parameter scarcity). Inter-model Top-K avoids this imbalance by deciding, at each coordinate, which tasks are allowed to contribute.
>
> To support this, we analyzed the first five GLUE tasks on GPT-2 (CoLA, MNLI, MRPC, QNLI, QQP). Using the same keep ratio (60%), intra-model Top-K + sign election (as in TIES) left 5.89% of parameters with no surviving task update. Our sign election + inter-model Top-K  reduced this to 2.06%. Thus, inter-model Top-K better matches the per-coordinate merging problem and more effectively utilizes the available task information.
> ## Question - 3
> > Have you investigated the interplay between the keep-ratio K and scaling factor λ ? For instance, does a more aggressive pruning (smaller K) necessitate a larger scaling factor λ to compensate? A 2D sensitivity plot for these two parameters would be insightful.
>
> We thank the reviewer for this insightful question. In response, we have added a 2D sensitivity plot of the K and λ in Figure 4 of the revision. The reviewer’s intuition about a correlation between the two hyperparameters is correct, although the effect we observe is opposite. When K is smaller (i.e., pruning becomes more aggressive), the remaining coordinates correspond to the largest task-vector magnitudes, which causes the Fisher-weighted median to increase at those coordinates. As a result, the merged model already moves sufficiently far from the base model, and therefore requires smaller scaling factors λ, not larger ones. Conversely, when K is larger and more low-magnitude deltas are retained, the median becomes more conservative, and a slightly larger λ helps reintroduce task-specific signal. We have highlighted this relationship clearly in Figure 4, and we appreciate the reviewer’s suggestion to visualize this interaction.
> ## Question - 4
> > While the method shows strong average performance, a qualitative analysis of its limitations would be valuable. For instance, the paper reports performance degradation on certain tasks (e.g., Multilingual in Table 2, Maths in Table 3). Could the authors provide an analysis of these failure cases? What might such an analysis reveal about the method's underlying assumptions or potential weaknesses?
>
> We thank the reviewer for this question. Variation in per-task PRR during ablations is a well-known phenomenon in the model-merging literature. For instance, TIES-Merging often performs below Task Arithmetic, Fisher merging, or RegMean on several individual tasks, yet achieves a stronger average performance by preventing extremely high gains on a few tasks from coinciding with substantial drops on others. Our ablations exhibit a similar pattern: removing components such as sign resolution may improve results on specific tasks like Maths or Multilingual, but DRIFT-MEDIAN reliably achieves higher mean PRR. This reflects a central goal of merging methods—to obtain a model that behaves robustly across the full set of tasks rather than prioritizing a small subset at the expense of overall consistency.

---

> ### Author Response · Authors · 2025-11-21
>
> ## Question - 5
> >The paper provides limited detail regarding the computation of the Fisher information matrix, which is a critical component. Could you please clarify the following:
> >- Scalability: How does the computational cost of this step scale with model size? Is it linear, or does it pose a significant bottleneck for even larger models
>
>
> >- Sample Selection: What criteria were used to select the samples for this computation? What is the composition of the "validation data" mentioned in line 978, how was it constructed, and is it fully disjoint from the test sets used for final evaluation?
>
>
> >- Domain Sensitivity: Have you analyzed how the domain coverage and diversity of the dataset used for Fisher estimation affect the final merging performance? For example, would using a more general-purpose dataset instead of a task-specific one alter the results?
>
> Our method computes a diagonal Fisher information matrix, and the computational cost linear w.r.t. model size and therefore fully scalable to larger LLMs.
> For LLM experiments, we use all available validation samples for each task without any subsampling or filtering. For vision tasks, we use 256 randomly selected samples from the validation split. Although more targeted sample selection could potentially improve Fisher estimates, we did not explore this direction. Importantly, the validation data used for Fisher estimation is always fully disjoint from the test sets, in case of GPT-2, where test set of GLUE benchmark does not provide public test labels, so as followed by previous research, we report results on the validation set while using randomly selected 256 examples from train set for fisher computation.
> We agree that the choice of data used for Fisher estimation can influence the merged model. In the paper, we compute Fisher using task-specific validation data for each domain, but we have not yet explored mismatched/general-purpose datasets. We acknowledge this as a limitation and note that analyzing domain sensitivity is an interesting direction for future work.

---

> ### Comment · Reviewer_8UxF · 2025-11-23
>
> I thank the authors for their detailed response and i have raised my score. However, some key concerns remain insufficiently addressed.
>
> > **1. Regarding Question 4**
>
> The authors state that a higher $K$ pulls the merged parameter back toward the pre-trained initialization, thus requiring a larger $\lambda$.  It is not theoretically clear why including more same-sign updates would dilute the update.
>
> > **2. Regarding Question 5**
>
> Since the primary contribution of this work is using Fisher information to measure parameter importance, the influence of the data used to compute the Fisher matrix is a key point of the method's effectiveness. Merely acknowledging the lack of analysis on data selection as a limitation is insufficient.

---

> > ### Author Response · Authors · 2025-11-27
> >
> > Thank you for going through the responses and also for raising the score. We would like to clarify the additional points.
> >
> > > The authors state that a higher pulls the merged parameter back toward the pre-trained initialization, thus requiring a larger . It is not theoretically clear why including more same-sign updates would dilute the update.
> >
> > Thank you for the opportunity to elaborate this.
> >
> >
> > Let the retained task updates at a given coordinate be numbers  $\tau_1, \tau_2, \ldots, \tau_M$  with non-negative Fisher weights  $F_1, F_2, \ldots, F_M$.
> > We define the total Fisher weight as: $W = \sum_{m=1}^M F_m$
> >
> > It is characterized by the condition that the cumulative Fisher weight on each side of $\tau^\*$ is at most half the total:
> >
> > $$
> > \sum_{\tau_m \le \tau^\*} F_m \ge \frac{W}{2},
> > \qquad
> > \sum_{\tau_m \ge \tau^\*} F_m \ge \frac{W}{2}.
> > $$
> >
> > When we increase the Top-$K$ value, many additional task updates with very small magnitude
> > (i.e., $\tau_m \approx 0$) enter the set.  Even though these updates have the correct sign, they cluster near zero.
> >
> >
> > As more near-zero updates accumulate with low-value $\tau$, this soon exceeds $W/2$, meaning the median conditions above are satisfied. Thus the Fisher-weighted median is mathematically forced toward the pretrained parameter, not because of sign disagreement but because *the median depends on cumulative weight*.
> >
> > In simpler words, assuming a toy-example with 5 models. Suppose we have five retained updates: 0.0, 0.02, 0.05, 0.6, and 0.7, all with equal Fisher weights for simplicity. If we use a stricter Top-K and prune the first two tiny updates, the remaining values are 0.05, 0.6, and 0.7, so the median is 0.6, capturing reasonable deviation from base. But if we increase K and keep all five values, then most of the weight now lies near 0.0, 0.02, and 0.05, so the median becomes 0.05 instead of 0.6.
> >
> > > Since the primary contribution of this work is using Fisher information to measure parameter importance, the influence of the data used to compute the Fisher matrix is a key point of the method's effectiveness. Merely acknowledging the lack of analysis on data selection as a limitation is insufficient.
> >
> > We would like to thank you for encouraging us to further analyze the domain senstivity for Fisher estimation. We performed additional experiments on CLIP-based vision tasks to evaluate how sensitive DRIFT-MEDIAN is to domain-mismatch.
> >
> > | Validation Data                                   | SUN397 | CARS  | RESISC45 | Eurosat | SVHN  | GTSRB | MNIST | DTD   | Average | PRR   |
> > |---------------------------------------------------|--------|-------|----------|---------|-------|-------|-------|-------|---------|-------|
> > | Unchanged                                         | 65.01  | 66.17 | 71.38    | 76.19   | 88.05 | 64.33 | 97.55 | 58.35 | 73.38   | 81.22 |
> > | MNIST → KMNIST                                    | 65.17  | 66.45 | 71.11    | 75.96   | 86.83 | 64.31 | 98.06 | 58.09 | 73.25   | 81.09 |
> > | MNIST → KMNIST and SVHN → MNIST                   | 65.33  | 66.92 | 71.98    | 77.19   | 77.43 | 64.65 | 97.36 | 58.35 | 72.40   | 80.25 |
> >
> > These results indicate that replacing MNIST with KMNIST: a substantial visual shift involving a different writing system (English to Japanese) produces only a minor change in overall performance, suggesting that DRIFT-MEDIAN tolerates moderate domain mismatch. In contrast, replacing SVHN (RGB street-view numbers with background clutter) with MNIST (grayscale digits) along with MNIST to KMIST causes a clear drop on SVHN but leaves the remaining domains largely unaffected.
> >
> > Beyond vision tasks, similar domain-mismatch effects naturally arise in the LLM setting. In fact, for Llama-3.1-8B and Llama-3.2-3B, Fisher estimation is performed using multilingual instruction-following data, whereas evaluation spans tasks such as ARC, HellaSwag, MMLU. Moreover, for tasks such as MBPP, HumanEval, and GSM8K, no official validation sets exist, so we rely on broadly related but distributionally distinct data. Despite this mismatch, DRIFT-MEDIAN consistently maintains strong performance across all LLM benchmarks, indicating that the method does not require tightly aligned Fisher estimation data to be effective.
> >
> > In summary, these experiments show that DRIFT-MEDIAN is generally resilient to reasonable domain shifts in the Fisher estimation stage and performs reliably even when the Fisher data and evaluation data come from different distributions. Only extreme mismatches, such as substituting SVHN with MNIST lead to noticeable degradation. However, it should be noted that, in such cases, the degradation on such tasks is also very high (88.05 to 77.43), and many other baselines perform better on that task. We have added these details in Appendix I.

---

### Official Review · Reviewer_nTuV · 2025-11-01

**Soundness:** 2
**Presentation:** 3
**Contribution:** 2
**Rating:** 4
**Confidence:** 4

**Summary:**

This paper proposes a task-aware model merging method that aims to enhance the robustness and adaptability of parameter-space merging across diverse tasks. The approach can be viewed as a parameter-level weighted merging strategy that, similar to FisherMerging, leverages task-specific data characteristics to guide the parameter combination process. Conceptually, the overall workflow can be regarded as a linear combination of TiesMerging and FisherMerging: it integrates the conflict-mitigation mechanism from TiesMerging with the parameter-sensitivity estimation from FisherMerging. A distinctive feature of the proposed method is the introduction of a cross-model redundancy pruning strategy, which differs from TiesMerging that performs pruning within a single model. The paper further explores several design choices—such as the use of the Fisher matrix, sign resolution, and top-K pruning—and evaluates the method on multiple tasks, including Natural Language Understanding, mathematics, and coding benchmarks. Experimental results demonstrate that the method can effectively improve merged model performance in many cases.

**Strengths:**

+ The paper provides a clear motivation for addressing the limitations of current parameter-space merging methods and positions its work within the context of task interference and parameter sensitivity.

+ By integrating the principles of TiesMerging and FisherMerging, the proposed approach attempts to balance parameter alignment and task sensitivity, offering a new perspective for parameter-space model merging.

+ The introduction of a cross-model redundancy pruning strategy represents an interesting attempt to reduce interference between tasks, distinguishing this method from prior single-model pruning approaches.

+ The paper follows a clear methodological pipeline and conducts extensive experiments across various task domains.

+ The method demonstrates adaptability across multiple types of tasks, suggesting its potential applicability to broader multi-task or multi-domain model merging scenarios, especially in generative tasks.

**Weaknesses:**

- The abstract claims that the proposed method bridges the gap between interference-handling approaches and parameter-sensitivity-aware approaches. However, since the two processes are combined in a serial manner, the method does not truly consider interference and parameter sensitivity simultaneously. These steps are independent. For example, important sensitive parameters may be pruned during the sign resolution step before the Fisher matrix is even computed.

- The proposed method is mainly derived from TiesMerging and FisherMerging, but it does not clearly state the specific improvements or innovations compared with these methods.

- The method is not data-free, as it still requires task data to compute the Fisher matrix. Therefore, it cannot be applied in completely data-free scenarios.

- The method workflow first computes the Fisher matrix and then performs cross-model pruning. However, if a parameter is highly sensitive for a given task but has only small update magnitudes due to the characteristics of that task, this parameter might be overshadowed by a less critical parameter from another task that has a larger update magnitude.

- The process of determining unified signs may already eliminate many offsets of task vectors at the corresponding parameter positions. The subsequent cross-model pruning step further removes parameters from some task vectors. As a result, performance might be dominated by a few tasks with large update magnitudes. This may be acceptable for NLP tasks, where tasks with small offsets can often be handled well by the base model itself. However, for vision-related tasks, where task offsets are generally larger, performance may not be as stable. It is therefore suggested to include more vision-domain tasks to verify this issue and further evaluate the cross-modal applicability of the proposed method.

- The key coefficient K for cross-model pruning is not thoroughly discussed in terms of how different values affect performance.

- The transition from Equation (2) to Equation (3) is not described in sufficient detail, although it is crucial for understanding the subsequent steps.

- Different hyperparameters are searched for each base model, and the reported results correspond to the optimal values found for each. However, according to the appendix, other baselines seem to use the same hyperparameters across different base models, which may make the comparison unfair. In addition, the claim that performance is insensitive to hyperparameter variations near the optimal value lacks theoretical or experimental support.

- The computation of the Fisher matrix requires significant additional computational resources and time, as well as extra storage space to retain the corresponding matrices.

- The paper lacks a discussion on how the performance of the proposed method changes as the number of tasks increases, which is crucial for verifying its ability to handle task conflicts and to effectively measure task-sensitive parameters.

- The work lacks comparisons with the latest methods. Although several recent approaches are introduced in the paper, they are not included in the experiments. For instance, PCB Merging appears only in Table 4 but is not included in other experiments. Moreover, AdaMerging (2024), which is also a task-aware method, is not compared.

- In the experimental section, most of the content focuses on reporting results, while deeper analysis and discussion of observed phenomena are missing. For example, the paper does not explain why the method performs worse on some tasks than simple task arithmetic.

- The paper lacks detailed descriptions of the specific task configurations and experimental setups used for the in-domain and out-of-domain experiments shown in Figure 3.

- The reference formatting is inconsistent. For instance, there is inconsistency in whether journal names are italicized, whether URLs and DOIs are provided, and in the abbreviation styles of journal or conference names. At the very least, papers from the same venue should follow a consistent citation format.

**Questions:**

Q1. In the first paragraph of the Introduction, it is stated that “current parameter-space merging methods exhibit fundamental limitations that constrain their effectiveness.” What exactly are these limitations?

Q2.The Fisher matrix already measures the sensitivity of parameters for a given task and reduces the influence of less important parameters. Why does the subsequent TOP-K selection use the magnitude of parameter updates as the criterion, instead of jointly considering parameter sensitivity? Could TOP-K possibly prune out parameters that are actually sensitive?

Q3. Compared with previous methods, this approach adds one more step of task vector decomposition and combination, making it a total of two steps. Is this necessary?

Q4. In Table 3, it can be observed that the proposed method performs worse than simple task arithmetic on mathematical and coding tasks. What is the reason for this?

Q5. In Table 4, the reproduced fine-tuned model performance is reported. During reproduction, were the parameters and environment exactly the same as those used in the first row of fine-tuning? Are the results in the last row obtained by merging your own fine-tuned models rather than those in the first row?

Q6. In the ablation study, after removing some key components such as sign resolution, the performance on some task types (e.g., Maths and Multilingual) even increases significantly. Why does this happen?

Q7. Why does the out-of-domain performance decrease as the merging coefficient increases? What might be the underlying reason for this trend?

Q8. In Figure 1(b), what does the symbol located between the two formulas at the bottom represent? It seems that this symbol is not used in the subsequent equations or steps.

Q9. For the scaling coefficient used in merging task vectors, what is the specific granularity used during the search?"

**Details Of Ethics Concerns:**

no ethical issues identified.

---

> ### Author Response · Authors · 2025-11-21
>
> ## Weakness - 1
> > The abstract claims that the proposed method bridges the gap between interference-handling approaches and parameter-sensitivity-aware approaches. However, since the two processes are combined in a serial manner, the method does not truly consider interference and parameter sensitivity simultaneously. These steps are independent. For example, important sensitive parameters may be pruned during the sign resolution step before the Fisher matrix is even computed.
>
> Although interference handling and Fisher weighting are applied sequentially, both forms of information jointly influence the final aggregation step, because the Fisher-weighted median is computed only over the parameters that survive interference resolution.
> Regarding the concern that sensitive parameters might be pruned, we acknowledge that this is possible. However, retaining such parameters would have adverse effects on tasks that disagree on their update directions. During the sign-resolution step, we determine the retained direction based on the aggregate sign across all task vectors; consequently, preserving a parameter whose direction is strongly disputed would amplify interference and degrade overall alignment. Our decision therefore reflects an explicit trade-off: we discard parameters whose directional inconsistency would harm multiple tasks in order to preserve those that contribute in same direction across tasks.
>
> ## Weakness - 2
> > The proposed method is mainly derived from TiesMerging and FisherMerging, but it does not clearly state the specific improvements or innovations compared with these methods.
>
> Thank you very much for raising this concern regarding novelty. We agree that, our method is derived from TIES Merging and Fisher merging. Our contribution, however, lies in unifying these two well-established disconnected approaches by integrating insights from both the approaches. One of the main challenges in parameter-space merging is keeping the right balance between the two  extremes: 1) remaining too close to the base model and 2) deviating too far from it. Both can  lead to suboptimal outcomes. If the merged parameters stay too close to the base model, they fail to incorporate the meaningful task-specific information contained in the fine-tuned task vectors, resulting in an under-specialized model. Conversely, if the merge shifts too far toward any individual task vector, the model may deviate from the pretrained manifold, often causing performance degradation on unrelated tasks. This is where Top-K and the Fisher-weighted median complement each other. Coordinate-wise Top-K selects the most meaningful task updates, preventing the merged parameters from being dominated by numerous small deltas. However, Top-K alone cannot guard against large but low-Fisher updates that may pull the merged parameters too far from the base model.
>
> Another important contribution is formulating closed-form solution based on the Fisher-weighted median, which ensures that parameter contributions capture both sensitivity and relevance, resulting in robust model merging. We have shown that closed form solution for weighted L1 loss can be represented in terms of Fisher weighted median.
>
>
>
> ## Weakness - 3
> > The method is not data-free, as it still requires task data to compute the Fisher matrix. Therefore, it cannot be applied in completely data-free scenarios.
>
> Yes, we completely agree with this. Although this is one of the limitation of our method, even a very small number of examples is sufficient. For instance, we use only 256 examples for GPT-2 and vision tasks. This is the default no. of examples used by previous works so we restrict ourselves with the same.

---

> ### Author Response · Authors · 2025-11-21
>
> ## Weakness - 4
> > The method workflow first computes the Fisher matrix and then performs cross-model pruning. However, if a parameter is highly sensitive for a given task but has only small update magnitudes due to the characteristics of that task, this parameter might be overshadowed by a less critical parameter from another task that has a larger update magnitude.
>
> We thank the reviewer for pointing this out. We agree that this situation is possible: a parameter that is Fisher-sensitive but exhibits a very small task-vector magnitude could, in principle, be overshadowed during the Top-K stage by a larger but less important update from another task. In practice, such parameters remain very close to the base model and thus have limited effect on the final Fisher-weighted median, but we acknowledge that this is a valid edge case. Importantly, it can be addressed by applying intra-model normalization prior to Top-K selection to ensure that inherently small yet meaningful deltas are not suppressed by large-magnitude spikes from unrelated tasks.
>
> We also examined whether this overshadowing effect correlates with the task-vector distance itself, but, as shown in Fig. 6 of the updated manuscript, we did not observe a clear or consistent relationship between task-vector norms and performance gaps. This suggests that the phenomenon is more nuanced and not simply governed by vector magnitude but interactions in the effective parameter space (Wang et al., 2025).
>
> ## Weakness - 5
> > The process of determining unified signs may already eliminate many offsets of task vectors at the corresponding parameter positions. The subsequent cross-model pruning step further removes parameters from some task vectors. As a result, performance might be dominated by a few tasks with large update magnitudes. This may be acceptable for NLP tasks, where tasks with small offsets can often be handled well by the base model itself. However, for vision-related tasks, where task offsets are generally larger, performance may not be as stable. It is therefore suggested to include more vision-domain tasks to verify this issue and further evaluate the cross-modal applicability of the proposed method.
>
> We thank the reviewer for raising this concern. We agree that sign resolution followed by Top-K pruning could, bias the merge toward tasks with larger updates. To evaluate this, we additionally ran DRIFT-MEDIAN on eight vision tasks using openai/clip-vit-base-patch32 as the base model, and observed improvements indicating that the method remains stable and effective beyond NLP. A intra-model normalization step can be added (similar to PCB merging) to handle such issues. We have included these results in the revised manuscript.
>
> ## Weakness - 6
> > The key coefficient K for cross-model pruning is not thoroughly discussed in terms of how different values affect performance.
>
> We thank the reviewer for raising this point. A larger value of K allows a greater number of task updates to be retained at each coordinate, including many low-magnitude deltas that lie close to the base model. They pull the Fisher-weighted median toward the pretrained parameters, reducing task-specific movement. To counteract this effect, the scaling coefficient λ needs to be increased so that meaningful task deviations maintain sufficient influence during aggregation. This interaction between K and λ is illustrated in Fig. 4 of the updated manuscript.
> ## Weakness - 7
> > The transition from Equation (2) to Equation (3) is not described in sufficient detail, although it is crucial for understanding the subsequent steps.
>
> Yes, we completely agree with the reviewer. Equation 3 can be derived from the Equation \hat{θ(m)} = θ^{(0)} + \hat{τ(m)} and Equation (2). We should have clearly mentioned that in the paper. However, we have clarified this in the revised manuscript .

---

> ### Author Response · Authors · 2025-11-21
>
> ## Weakness - 8
> > Different hyperparameters are searched for each base model, and the reported results correspond to the optimal values found for each. However, according to the appendix, other baselines seem to use the same hyperparameters across different base models, which may make the comparison unfair. In addition, the claim that performance is insensitive to hyperparameter variations near the optimal value lacks theoretical or experimental support.
>
> We thank the reviewer for this observation. For fairness, we did perform detailed hyperparameter searches on GPT-2 tasks and the vision experiments, including λ values in {0.1, 0.2, …, 1.0} and K in {0.3, 0.4, 0.5, 0.6, 0.7} for TIES, and comparable λ values for Task Arithmetic and other baselines; in all cases, our method outperformed the best configuration of these baselines. For the LLM experiments, we followed the hyperparameters reported in prior work (MergeBench), but we also tested small deviations around those settings and found they generally produced lower performance; we acknowledge that a full hyperparameter search over the entire space has not yet been performed, and we will include this in future version of the paper. Regarding the claim that performance is relatively insensitive near the optimal region, we have added a performance–hyperparameter sensitivity plot to empirically support this statement.
>
> ## Weakness - 9
> > The computation of the Fisher matrix requires significant additional computational resources and time, as well as extra storage space to retain the corresponding matrices.
>
>
> While Fisher estimation does introduce extra computation, its cost is modest in practice because it consists only of forward and backward passes without any optimizer updates, these are identical to steps already used during training. The ~1 hour per domain reported for an 8B model is primarily due to (a) the relatively long sequence lengths in the validation sets we use and (b) our current implementation running with a small batch size (4) on a single GPU, as we have not parallelized Fisher estimation across multiple GPUs. Storage overhead is also manageable as we retain only the diagonal Fisher, which is a single parameter-sized vector. We will clarify in the revision that although Fisher estimation is an additional step, both its computational and memory requirements fit comfortably within routine large-model training workflows.
>
> ## Weakness - 10
> > The paper lacks a discussion on how the performance of the proposed method changes as the number of tasks increases, which is crucial for verifying its ability to handle task conflicts and to effectively measure task-sensitive parameters.
>
> We thank the reviewer for raising this important question. We agree that understanding how performance evolves as the number of merged tasks increases is crucial for assessing a method's robustness to task conflicts and its ability to identify task-sensitive parameters. However, we note that this behavior does not depend solely on the number of tasks; it is jointly influenced by several additional factors such as task similarity, parameter-space overlap, update sparsity, and the inherent variance activated by each task. As highlighted in Wang et al. (2025), the effective parameter space saturates as more experts are merged, driven by Gaussian Width concavity and redundancy constraints. The performance may plateau or even degrade depending on these structural properties rather than task count alone. While we fully recognize the importance of this question, a proper treatment requires a dedicated and carefully controlled study of these interacting factors, which is beyond the scope of the current paper. We have added a discussion on this in the appendix.

---

> ### Author Response · Authors · 2025-11-21
>
> ## Weakness - 11 and Question - 5
> > The work lacks comparisons with the latest methods. Although several recent approaches are introduced in the paper, they are not included in the experiments. For instance, PCB Merging appears only in Table 4 but is not included in other experiments. Moreover, AdaMerging (2024), which is also a task-aware method, is not compared.
> > In Table 4, the reproduced fine-tuned model performance is reported. During reproduction, were the parameters and environment exactly the same as those used in the first row of fine-tuning? Are the results in the last row obtained by merging your own fine-tuned models rather than those in the first row?
>
> We thank the reviewer for highlighting this point. For PCB Merging, we could not find an implementation for LLMs in the offical codebase and exact evaluation setup used for their reported results is unclear. For this reason, we were unable to obtain PCB results for the Llama-3B and Llama-8B settings. However, we included PCB in Table 4 by reproducing the scores of same fine-tuned models as theirs (checkpoints are publicly available) and compared directly to the values reported in their paper. To partially address this gap, we ported the PCB algorithm into our codebase for the vision experiments, where the required implementation details were clearer, and we include those comparisons. Similarly, we incorporated AdaMerging (2024) into our vision-task evaluation. Our method outperforms both PCB Merging and AdaMerging.
>
> ## Weakness - 12, Question - 4 and Question - 6
>
>
> > In the experimental section, most of the content focuses on reporting results, while deeper analysis and discussion of observed phenomena are missing. For example, the paper does not explain why the method performs worse on some tasks than simple task arithmetic.
> > In Table 3, it can be observed that the proposed method performs worse than simple task arithmetic on mathematical and coding tasks. What is the reason for this?
> > In the ablation study, after removing some key components such as sign resolution, the performance on some task types (e.g., Maths and Multilingual) even increases significantly. Why does this happen?
>
> We thank the reviewer for this question. Per-task PRR variance in ablations and baselines is expected and has been documented in the model-merging literature: for example, TIES-Merging underperforms Task Arithmetic, Fisher merging, and RegMean on several individual tasks, yet achieves a higher average performance by avoiding extremely high scores on a few tasks at the cost of severe degradation on others. Our ablations show the same behavior: removing components like sign resolution can occasionally boost performance on specific tasks such as Maths or Multilingual, but DRIFT-MEDIAN consistently improves mean PRR. This reflects the core design principle of merging methods: to produce a model that performs robustly  across tasks, rather than excelling on a narrow subset at the expense of overall balance.
> ## Weakness - 13
> > The paper lacks detailed descriptions of the specific task configurations and experimental setups used for the in-domain and out-of-domain experiments shown in Figure 3.
>
> Thanks for pointing this out. We have incorporated the details of experimental setup and task configuration in the respective section of the manuscript.
>
>
> ## Weakness - 14
> > The reference formatting is inconsistent. For instance, there is inconsistency in whether journal names are italicized, whether URLs and DOIs are provided, and in the abbreviation styles of journal or conference names. At the very least, papers from the same venue should follow a consistent citation format.
>
>
> Most of these discrepancies arose from papers sourced from arxiv. The bibtex format on arxiv was updated recently, and entries downloaded before a certain date did not include fields such as the URL, which led to inconsistent formatting across references. We have now used the latest bibtex provided on arxiv. For all other papers, we used the references from the official bibtex supplied by the respective publishers or venues.

---

> ### Author Response · Authors · 2025-11-21
>
> ## Question - 1
> > In the first paragraph of the Introduction, it is stated that “current parameter-space merging methods exhibit fundamental limitations that constrain their effectiveness.” What exactly are these limitations?
>
> We thank the reviewer for pointing this out. We have rephrased the corresponding paragraph in the Introduction to better reflect our intended meaning. Our intention was to suggest that prior merging methods primarily operate within a single paradigm: either interference-handling or parameter-sensitivity awareness, without jointly considering both perspectives within the same merging pipeline.
> ## Question - 2
> > The Fisher matrix already measures the sensitivity of parameters for a given task and reduces the influence of less important parameters. Why does the subsequent TOP-K selection use the magnitude of parameter updates as the criterion, instead of jointly considering parameter sensitivity? Could TOP-K possibly prune out parameters that are actually sensitive?
>
> We thank the reviewer for raising this point. While it is possible for a Fisher-sensitive parameter to have a very small task-vector update and thus be pruned by Top-K, such cases indicate that the task does not require meaningful deviation from the base model at that coordinate. In other words, the parameter is important, but the pretrained value is already suitable, so there is little task-specific signal to preserve. Our goal is precisely not to prioritize Fisher importance unless it is accompanied by task-specific information. For a task to receive high priority at a coordinate, it must contribute both a non-trivial update magnitude and high Fisher importance; only such parameters survive Top-K and meaningfully influence the weighted-median aggregation.
> ## Question - 3
> > Compared with previous methods, this approach adds one more step of task vector decomposition and combination, making it a total of two steps. Is this necessary?
>
> Yes, the additional step of task-vector decomposition and recombination is necessary. Fisher information is computed after sign resolution, because Fisher is evaluated on (base model + task vector). The rationale for performing sign resolution early is that we want the Fisher matrix to reflect a parameter space that is already aligned with the eventual merged model. Ideally, Fisher should be computed after all sparsification and scaling (Lee et al., 2025), but doing so would require recomputing the Fisher matrix for every hyperparameter setting, which is computationally infeasible. We therefore compute Fisher once, after sign resolution (which does not depend on hyperparameters), and apply Top-K and λ after Fisher to avoid repeated Fisher estimation. This is not a bottleneck, the decomposition and recomposition complete in under a minute even for 8B parameter models.
>
>
>
> ## Question - 7
> > Why does the out-of-domain performance decrease as the merging coefficient increases? What might be the underlying reason for this trend?
>
> Out-of-domain performance declines as the merging coefficient increases because a larger coefficient amplifies the influence of task vectors, which encode task-specific knowledge. These shifts override portions of the pretrained base model parameters, which are responsible for broad, domain-agnostic generalization. As more weight is placed on task-specific updates, the model becomes over-aligned with the fine-tuning distributions and progressively loses the representational flexibility required to handle unseen inputs. Since task vectors assume a meaningful base to modify, excessively large coefficients distort this relationship, eventually degrading both generalization and even in-domain specialization, consistent with the behavior shown in Figure 3.
>
> ## Question - 8
> > In Figure 1(b), what does the symbol located between the two formulas at the bottom represent? It seems that this symbol is not used in the subsequent equations or steps.
>
> Aplogies for the oversight. The symbol in Figure 1(b) was intended to denote the overall resolved sign s at that coordinate (not gamma, we switched the notations later on). We have corrected the figure. We thank the reviewer for pointing this out.
> ## Question - 9
> > For the scaling coefficient used in merging task vectors, what is the specific granularity used during the search?"
>
> We searched with increments of 0.05 starting from 1.1 to 1.5, we have added these details in appendix E.

---

> ### Author Response · Authors · 2025-11-21
>
> **References**
> * Zijing Wang, Xingle Xu, Yongkang Liu, Yiqun Zhang, Peiqin Lin, Shi Feng, Xiaocui Yang, Daling Wang and Hinrich Schütze. “Why Do More Experts Fail? A Theoretical Analysis of Model Merging.” ArXiv abs/2505.21226 (2025): n. pag.
> * Sanwoo Lee, Jiahao Liu, Qifan Wang, Jingang Wang, Xunliang Cai, and Yunfang Wu. 2025. Dynamic Fisher-weighted Model Merging via Bayesian Optimization. In Proceedings of the 2025 Conference of the Nations of the Americas Chapter of the Association for Computational Linguistics: Human Language Technologies (Volume 1: Long Papers), pages 4923–4935, Albuquerque, New Mexico. Association for Computational Linguistics.

---

### Official Review · Reviewer_CUZC · 2025-11-01

**Soundness:** 3
**Presentation:** 3
**Contribution:** 2
**Rating:** 2
**Confidence:** 3

**Summary:**

This paper proposes DRIFT-MEDIAN, a model merging method that aims to unify two dominant paradigms in the field: techniques that mitigate parameter interference (e.g., sign conflicts and redundancy) and those that account for parameter sensitivity to task performance. The method first resolves task vector conflicts via sign alignment, then quantifies parameter importance using Fisher information. It subsequently introduces a coordinate-wise Top-K selection strategy to retain the most relevant task vectors across models, and finally aggregates them using a Fisher-weighted median, which has a closed-form solution. The authors conduct experiments on various large language models (Llama family, GPT-2) across a range of tasks, reporting that their method outperforms existing baselines.

**Strengths:**

1. The paper is well-motivated. It correctly identifies two important yet often disconnected lines of research in model merging—parameter interference resolution and sensitivity-based weighting—and sets a valuable goal of unifying them into a single framework.
2. The proposed DRIFT-MEDIAN integrates multiple components (sign resolution, Top-K selection, Fisher-weighted median aggregation) into a logically coherent pipeline where each step serves a clear purpose.
3. The paper assesses the method on multiple LLMs of varying scales and architectures, covering a diverse set of tasks including mathematics, coding, multilingual understanding, and safety, which provides a relatively comprehensive testbed for its effectiveness.

**Weaknesses:**

1. The major concern is the limited novelty. Its core components—sign resolution, Top-K selection, and Fisher weighting—are heavily inspired by or directly adopted from existing works like TIES-merging and Fisher-merging. While the contribution lies in their combination, the assembly feels somewhat straightforward and lacks a deeper, more innovative mechanism.

2. The motivation and advantage of the coordinate-wise Top-K selection are not well-substantiated. The paper claims its coordinate-wise (row-wise) selection is superior to TIES's model-wise (column-wise) approach at preventing "parameter crowding and scarcity," but this claim is primarily supported by a simple schematic (Figure 2) and lacks more rigorous theoretical analysis or empirical evidence to prove its necessity and superiority.

3. The experimental results are not consistently convincing. On certain key tasks, the proposed method performs worse than some baselines. For instance, in Table 2, its PRR on the Maths task (85.19) is significantly lower than TIES (96.44) and Task Arithmetic (93.85). Similarly, in Table 3 on the Maths task, its PRR (65.77) is also lower than Task Arithmetic (72.40) and DARE (70.34). These inconsistent outcomes weaken the central claim of "consistently outperforming" prior methods.

**Questions:**

Please see the weaknesses.

---

> ### Author Response · Authors · 2025-11-21
>
> ## Weakness - 1
> > The major concern is the limited novelty. Its core components—sign resolution, Top-K selection, and Fisher weighting—are heavily inspired by or directly adopted from existing works like TIES-merging and Fisher-merging. While the contribution lies in their combination, the assembly feels somewhat straightforward and lacks a deeper, more innovative mechanism.
>
> Thank you very much for raising this concern regarding novelty. We agree that the individual components in our method have appeared in prior work in different forms. Our contribution, however, is in proposing a single, coherent formulation in which they interact in a principled and mutually supportive way. To the best of our knowledge, our approach is the first to jointly account for three aspects that are usually treated separately: the intra-model importance reflected in the structure of task vectors, the inter-model competition that arises when many tasks attempt to update the same coordinates, and the parameter sensitivity captured by Fisher information. DRIFT-MEDIAN ties these together by casting merging at each coordinate as a Fisher-weighted L1 consensus problem, applied after interference has been resolved and parameter-wise competition has been regulated.
>
> This integrated design is not simply a matter of chaining familiar steps. The sign-resolved deltas create a stable foundation for meaningful Fisher-based aggregation; the inter-model Top-K selection limits how many tasks can influence a particular parameter; and the Fisher-weighted median provides a more robust and sensitivity-aware merged value than the mean. As shown in Table 6, replacing the median with a mean leads to a clear performance degradation, which illustrates why the particular combination of ingredients matters and why their interaction is central to the method.
>
> We also note that although a recent work (Sun et al., 2025) considers pruning a small percentage of conflicting parameters based on sensitivity, the merged values for those coordinates are essentially taken from the base model. This differs from our goal of retaining and robustly aggregating task-specific information even in regions where tasks disagree. Because our approach resolves interference first, and then applies Fisher-weighted robust aggregation rather than reverting to the base model, it preserves a larger portion of the useful signal carried by each task.
>
> We have revised the paper to make this conceptual integration clearer, and to express more explicitly how DRIFT-MEDIAN differs from simply assembling existing ideas.
>
> ## Weakness - 2
> > The motivation and advantage of the coordinate-wise Top-K selection are not well-substantiated. The paper claims its coordinate-wise (row-wise) selection is superior to TIES's model-wise (column-wise) approach at preventing "parameter crowding and scarcity," but this claim is primarily supported by a simple schematic (Figure 2) and lacks more rigorous theoretical analysis or empirical evidence to prove its necessity and superiority.
>
> Thank you for raising this important point. Our motivation for coordinate-wise Top-K is consistent with findings from Qu et al. (2025), which shows that model merging fundamentally reduces to a one-dimensional estimation problem per parameter. Since interference, variance, and estimator instability all arise per coordinate, the sparsification step should also operate per coordinate to remain aligned with the structure of the estimator. In contrast, model-wise Top-K performs intra-model sparsification: each model independently discards many coordinates without considering how many models concentrate their remaining mass on the same coordinates. This leads to multiple task vectors pushing large updates onto a small subset of parameters. Our coordinate-wise (i.e., inter-model) Top-K directly limits how many models may influence any given coordinate, enforcing balanced cross-task competition at the location where the merged model ultimately must produce a single final value. Related work such as PCB-Merging (Zhang et al., 2024) also emphasizes the importance of balancing intra-model and inter-model interactions, although PCB does not incorporate parameter sensitivity. Our method unifies Fisher-aware sensitivity with inter-model competition control. In the revision, we have replaced the "row-wise/column-wise" terminology with the more precise intra-model vs. inter-model Top-K terminology.

---

> ### Author Response · Authors · 2025-11-21
>
> (cont.)
> To further support this rationale, we conducted an analysis on the first five GLUE tasks using GPT-2 (CoLA, MNLI, MRPC, QNLI, QQP). With a keep ratio of 60%, intra-model Top-K + sign election (as in TIES) leaves 5.89% of parameters with no surviving task update—i.e., no task contributes at those coordinates, forcing a fallback to the base model (parameter scarcity). In contrast, sign election + inter-model Top-K reduces this to 2.06%, meaning far fewer coordinates are left unused. This demonstrates that inter-model Top-K more closely matches the per-coordinate merging objective, reduces update scarcity, and more effectively utilizes the available task information. We have added these details in Section 4.2 of the updated paper.
> ## Weakness - 3
>
> > The experimental results are not consistently convincing. On certain key tasks, the proposed method performs worse than some baselines. For instance, in Table 2, its PRR on the Maths task (85.19) is significantly lower than TIES (96.44) and Task Arithmetic (93.85). Similarly, in Table 3 on the Maths task, its PRR (65.77) is also lower than Task Arithmetic (72.40) and DARE (70.34). These inconsistent outcomes weaken the central claim of "consistently outperforming" prior methods.
>
> We agree that the current phrasing in the paper (consistently outperforming) could have been clearer. Our intention was to refer to mean PRR across tasks, not to every individual task independently. We have revised the wording in the papaer accordingly.
> It has been estabilished in model-merging literature that per-task PRR variance is normal and expected. For example, TIES-Merging (Yadav et al., 2023) underperforms Task Arithmetic, Fisher merging, and RegMean on several individual tasks, yet it achieves a superior average performance, avoiding extremely high performance on some tasks at the cost of severe degradation on others.
> Similarly, in our results:
> - DRIFT-MEDIAN improves the mean PRR across all tasks and settings, despite not achieving the best score on every individual task.
> - This is a core design principle of merging methods: to obtain a model that performs robustly and uniformly, rather than excelling in a small subset of tasks while failing on others.
> We have revised the claims in the paper.
>
>
> **References**
> - Prateek Yadav, Derek Tam, Leshem Choshen, Colin Raffel, and Mohit Bansal. 2023. TIES-MERGING: resolving interference when merging models. In Proceedings of the 37th International Conference on Neural Information Processing Systems (NIPS '23). Curran Associates Inc., Red Hook, NY, USA, Article 310, 7093–7115.
>
> - Xingyu Qu and Samuel Horváth. "Vanishing Feature: Diagnosing Model Merging and Beyond." The Second Conference on Parsimony and Learning (Proceedings Track).
>
> - Guodong Du, Junlin Lee, Jing Li, Runhua Jiang, Yifei Guo, Shuyang Yu, Hanting Liu, Sim Kuan Goh, Ho-Kin Tang, Daojing He, and Min Zhang. 2024. Parameter competition balancing for model merging. In Proceedings of the 38th International Conference on Neural Information Processing Systems (NIPS '24), Vol. 37. Curran Associates Inc., Red Hook, NY, USA, Article 2691, 84746–84776.
>
> - Wenju Sun, Qingyong Li, Wen Wang, Yangliao Geng, and Boyang Li. 2025. Task Arithmetic in Trust Region: A Training-Free Model Merging Approach to Navigate Knowledge Conflicts. In Proceedings of the 33rd ACM International Conference on Multimedia (MM '25). Association for Computing Machinery, New York, NY, USA, 5178–5187. https://doi.org/10.1145/3746027.3755789

---

> ### Comment · Reviewer_CUZC · 2025-11-27
>
> Thank you for the detailed response. Your clarifications address several of my concerns regarding the empirical results and methodological choices. However, my primary reservation about the novelty of the approach remains. Even with the additional explanations, the core contribution still feels incremental and does not, in my view, reach the level of originality typically expected for ICLR. Therefore, I will maintain my original score.

---

### Note · Authors · 2026-01-05

I have read and agree with the venue's withdrawal policy on behalf of myself and my co-authors.